# Compartmentalized metabolism supports midgestation mammalian development

Ashley Solmonson[1], Brandon Faubert[1,2], Wen Gu[1], Aparna Rao[1], Mitzy A. Cowdin[3], Ivan Menendez-Montes[4], Sherwin Kelekar[1], Thomas J. Rogers[1], Chunxiao Pan[1], Gerardo Guevara[1], Amy Tarangelo[1], Lauren G. Zacharias[1], Misty S. Martin-Sandoval[1], Duyen Do[1], Panayotis Pachnis[1], Dennis Dumesnil[1], Thomas P. Mathews[1], Alpaslan Tasdogan[1,5], An Pham[6], Ling Cai[1,7], Zhiyu Zhao[1], Min Ni[1], Ondine Cleaver[3], Hesham A. Sadek[3,4], Sean J. Morrison[1,8] & Ralph J. DeBerardinis[1,6,8] ✉

Mammalian embryogenesis requires rapid growth and proper metabolic regulation[1]. Midgestation features increasing oxygen and nutrient availability concomitant with fetal organ development[2,3]. Understanding how metabolism supports development requires approaches to observe metabolism directly in model organisms in utero. Here we used isotope tracing and metabolomics to identify evolving metabolic programmes in the placenta and embryo during midgestation in mice. These tissues differ metabolically throughout midgestation, but we pinpointed gestational days (GD) 10.5–11.5 as a transition period for both placenta and embryo. Isotope tracing revealed differences in carbohydrate metabolism between the tissues and rapid glucose-dependent purine synthesis, especially in the embryo. Glucose's contribution to the tricarboxylic acid (TCA) cycle rises throughout midgestation in the embryo but not in the placenta. By GD12.5, compartmentalized metabolic programmes are apparent within the embryo, including different nutrient contributions to the TCA cycle in different organs. To contextualize developmental anomalies associated with Mendelian metabolic defects, we analysed mice deficient in LIPT1, the enzyme that activates 2-ketoacid dehydrogenases related to the TCA cycle[4,5]. LIPT1 deficiency suppresses TCA cycle metabolism during the GD10.5–GD11.5 transition, perturbs brain, heart and erythrocyte development and leads to embryonic demise by GD11.5. These data document individualized metabolic programmes in developing organs in utero.

Metabolism supports tissue development by supplying metabolic intermediates for energy production, anabolism, epigenetic regulation of gene expression and the formation of metabolic gradients that inform embryonic patterning[6–8]. The post-implantation embryo and placenta initially develop in relative hypoxia[9] (1–5% $O_2$). During this period, both the placenta and embryo require hypoxia-inducible gene-expression programmes, and disrupting these pathways or prolonging exposure to hypoxia results in improper cell differentiation and premature lethality[10–12] around GD10. Midgestation is marked by an increased transfer of nutrients and oxygen from the maternal circulation as fetal erythropoiesis begins and the vasculature matures in the placenta and embryo. This period is characterized by accelerating growth of placenta and embryo, and morphogenesis in the heart, brain and liver[2,13] (Fig. 1a), both of which suggest that midgestation is a metabolically dynamic period. Genetic and environmental alterations of metabolism result in developmental defects in humans[14–16], although the mechanism of many such anomalies is unknown. Most previous analyses of mouse embryonic metabolism has relied on ex vivo models or inferred metabolic requirements indirectly from the developmental consequences of genetic loss-of-function experiments[1]. We set out to observe metabolism directly in the intact fetal–placental unit in vivo during midgestation to identify metabolic transitions and to test the effects of perturbing them.

## Distinct metabolic transitions at GD10.5

In mice, placentation begins[2] at GD3.5 and facile dissection of the placenta from the embryo is possible by GD9.5. To characterize metabolism during midgestation, we collected embryos and placentas from naively pregnant C57BL/6J dams from GD10.5 to GD13.5 and performed metabolomics. Tissue mass increased rapidly in both the placenta and embryo over this period (Fig. 1b). Placenta and embryo metabolomics differ throughout midgestation, as expected given their divergent cellular composition and functions (Fig. 1c, Extended Data Fig. 1).

[1]Children's Medical Center Research Institute, University of Texas Southwestern Medical Center, Dallas, TX, USA. [2]Section of Hematology and Oncology, Department of Medicine, The University of Chicago, Chicago, IL, USA. [3]Department of Molecular Biology, University of Texas Southwestern Medical Center, Dallas, TX, USA. [4]Division of Cardiology, Department of Internal Medicine, University of Texas Southwestern Medical Center, Dallas, TX, USA. [5]Department of Dermatology, University Hospital Essen and German Cancer Consortium, Partner Site Essen, Essen, Germany. [6]Department of Pediatrics, University of Texas Southwestern Medical Center, Dallas, TX, USA. [7]Quantitative Biomedical Research Center, Department of Population and Data Sciences, University of Texas Southwestern Medical Center, Dallas, TX, USA. [8]Howard Hughes Medical Institute, University of Texas Southwestern Medical Center, Dallas, TX, USA. ✉e-mail: ralph.deberardinis@utsouthwestern.edu

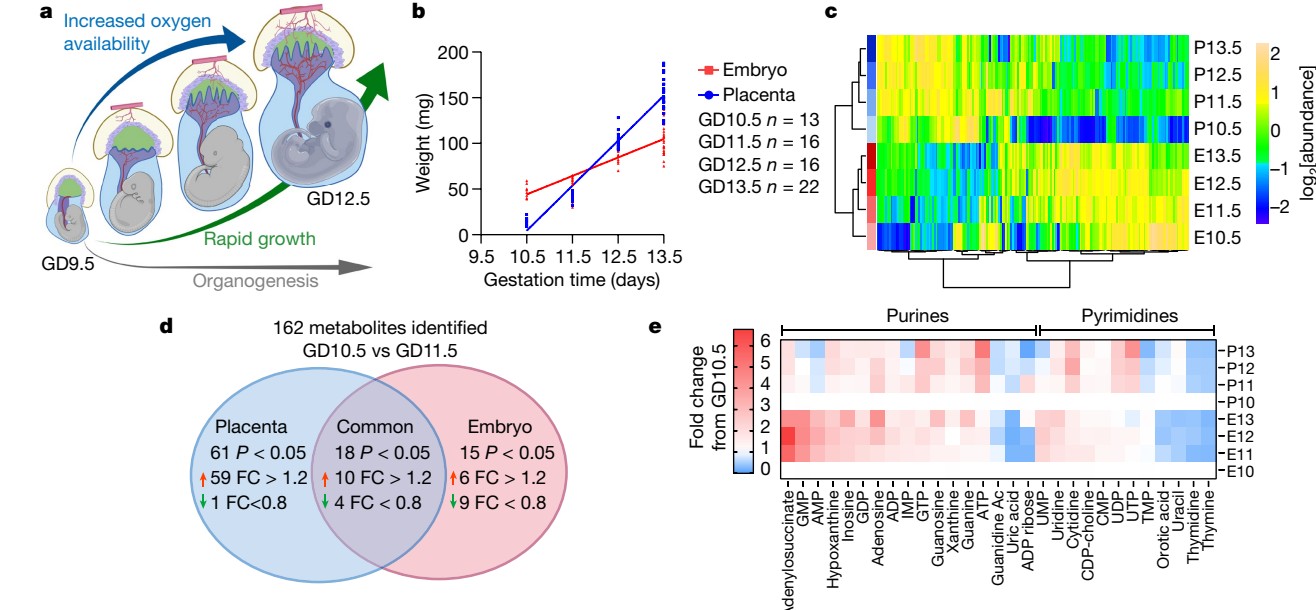

**Fig. 1 | Metabolic transition at GD10.5–GD11.5. a**, Midgestation is a dynamic period of development. **b**, Tissue weights from pregnant dams, aged 13.6 ± 3.8 weeks. **c**, Group average heat map of metabolomics data. **d**, Metabolites with $P < 0.05$ and fold change (FC) > 1.2 or < 0.8 between GD10.5 and GD11.5. **e**, Heat map of purines and pyrimidines, plotted as fold change relative to GD10.5.

Statistical tests: straight-line least-squares fitting followed by the extra sum-of-squares $F$-test (**b**); Student's $t$-tests (**d**). Data are mean ± s.d. Statistical tests were two-sided. Guanidine Ac, guanidine acetate; IMP, inosine monophosphate (additional abbreviations, Supplementary Table 1).

In both tissues, GD10.5 was metabolically different from subsequent days, indicating transitions between GD10.5 and GD11.5 (Fig. 1c, Extended Data Fig. 1). These transitions were largely distinct between embryo and placenta, with most metabolites changing in one tissue but not the other (Fig. 1d, Extended Data Fig. 2a–d). Metabolic set overrepresentation analysis (MSOA) identified numerous pathways that change in the placenta between GD10.5 and GD11.5, particularly pathways related to nitrogen and amino acid metabolism (Extended Data Fig. 2d). Urea cycle-related metabolites increased abruptly but transiently in the placenta at GD11.5 (Extended Data Fig. 2e), possibly reflecting the role of arginine in stimulating placental–fetal blood flow[17]. In the embryo, MSOA between GD10.5 and GD11.5 identified purine and pyrimidine metabolism as two of the top-scoring pathways (Extended Data Fig. 2c). Most purines displayed a sustained increase after GD10.5 in the embryo, whereas pyrimidines showed little change or decreased in both tissues (Fig. 1e).

## Rapid and localized metabolism in utero

To assess metabolite turnover in utero, we adapted previous methods[18] to infuse uniformly labelled [$^{13}$C]-glucose ([U-$^{13}$C]glucose) into pregnant mice at GD10.5. Embryos and adjoined placentas were removed every 30 min while uterine blood flow was maintained so that nutrient transport and metabolism could be assessed kinetically (Fig. 2a). This analysis revealed rapid labelling in maternal blood, placenta and embryo, indicating efficient glucose transfer from maternal circulation to embryo, as expected (Fig. 2b). By contrast, embryonic glutamine was labelled slowly from [U-$^{13}$C]glutamine in the maternal circulation (Fig. 2c), indicating distinct transport kinetics for different nutrients.

Rapid labelling of downstream metabolites indicates robust metabolism in the conceptus, and distinct labelling features in the embryo and placenta indicate metabolic differences between the tissues. Levels of $^{13}$C enrichment in glucose-derived metabolites reflect the combined contribution of labelled and unlabelled substrates through intersecting pathways (Extended Data Fig. 3a). Glucose-6-phosphate (G6P) appeared rapidly in the embryo as m+6, indicating conversion from maternal

glucose (Fig. 2d). However, placental G6P was labelled differently in the same mice. Overall G6P enrichment was lower than in the embryo, and G6P m+6 and m+3 appeared over similar time scales (Fig. 2d). A complete understanding of carbohydrate metabolism will require compartment-specific enzyme knockouts, but the placental labelling pattern suggests contributions from glycogenolysis, gluconeogenesis and other pathways previously reported in mammalian placentas[19] (Extended Data Fig. 3a).

The pentose phosphate pathway intermediate and nucleotide precursor ribose-5-phophate (R5P) also turned over rapidly. R5P was similar to G6P in that labelling was distributed across several isotopologues, and fully labelled R5P (m+5) was the predominant labelled form in the embryo but not the placenta (Extended Data Fig. 3b, c). After 4 h, purines were extensively labelled in both placenta and embryo, but again labelling was higher in embryos (Fig. 2e, Extended Data Fig. 3d–f). The total enrichment (that is, 1.0 − the unlabelled fraction, incorporating all $^{13}$C-labelled forms) was above 0.3 in all purines analysed, indicating that within 4 h, at least 30% of the embryo purine pools contained carbon originating in the maternal circulation (Fig. 2e). Although much of the purine labelling appeared to arise from R5P, purine bases in the embryo also contained $^{13}$C; evidence for labelling in the bases included higher total labelling in purines than R5P (Fig. 2e), and the presence of inosine monophosphate, GMP and AMP containing more than five $^{13}$C nuclei (Extended Data Fig. 3d–f). In the context of the expanding purine pool (Fig. 1e) and extensive labelling of serine and glycine (Extended Data Fig. 3g), these data point to de novo purine synthesis in embryos. As an orthogonal labelling approach, we infused pregnant mice with [γ-$^{15}$N]glutamine. The labelled nitrogen is incorporated into the purine ring during de novo synthesis. Again, higher relative enrichments were detected in inosine monophosphate and GMP in the embryos (Fig. 2f). Pyrimidines were also labelled by both [U-$^{13}$C]glucose and [γ-$^{15}$N]glutamine, but with less consistent differences between embryo and placenta (Extended Data Fig. 3h, i). Overall, the data indicate rapid metabolism during midgestation, including prominent utilization of maternal glucose and glutamine for embryonic purines, and distinct patterns of metabolic labelling between embryo and placenta.

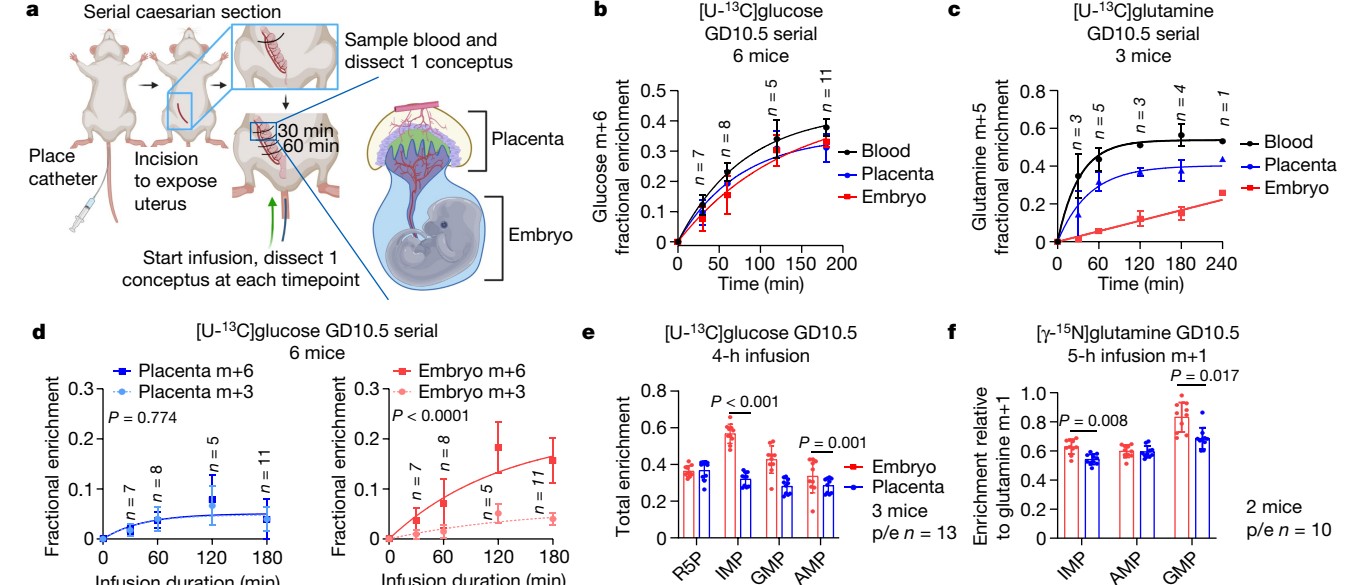

**Fig. 2 | Carbohydrate metabolism in midgestation. a**, Serial caesarian-section procedure. **b**, **c**, Time-dependent enrichment of [U-13C] glucose (**b**) and [U-13C]glutamine (**c**). **d**, Major glucose-6-phosphate isotopologues during serial caesarian-section infusion. **e**, Total enrichment (1 – unlabelled) of purines from [U-13C]glucose. **f**, M+1 enrichment in purines from [γ-15N]glutamine. 15N-glutamine enrichments are normalized to glutamine m+1 to account for differences among compartments (see Fig. 2c). Statistical tests: plateau followed by one-phase decay least-squares fitting followed by the Holm–Sidak's multiple-comparisons adjustment (**b**–**d**) (**b**: embryo vs placenta $P = 0.09$, embryo vs blood $P = 0.0001$, placenta vs blood $P = 0.003$; **c**: embryo vs placenta $P < 0.0001$, embryo vs blood $P < 0.0001$, placenta vs blood $P < 0.0001$); paired $t$-tests or Wilcoxon matched-pairs signed-rank tests followed by Holm–Sidak's multiple-comparisons adjustment (**e**); $\log_2$ paired $t$-tests followed by Holm–Sidak's multiple-comparisons adjustment (**f**). Data are mean ± s.d. Statistical tests were two-sided. p/e, placenta/embryo.

## Compartmentalized embryonic metabolism

Increases in vascularization, erythropoiesis and cardiac function[2,20] predict enhanced oxidative metabolism in the embryo during midgestation. We performed [U-13C]glucose infusions between GD9.5 and GD12.5, when oxygen levels increase[9]. On GD9.5, the placenta displayed higher labelling of tricarboxylic acid (TCA) cycle intermediates than the embryo (Fig. 3a). Labelling in the placental TCA cycle intermediates changed minimally over the next 3 days, but labelling in embryonic intermediates increased such that by GD12.5, labelling was similar or higher in the embryo than the placenta (Fig. 3b, Extended Data Fig. 4a, b). The citrate m+2/pyruvate m+3 ratio reports transfer of labelled two-carbon units from glucose to citrate via pyruvate dehydrogenase (PDH), whereas the citrate m+3/pyruvate m+3 ratio reports transfer of labelled three-carbon units via pyruvate carboxylase. In both tissues on all days, citrate m+2/pyruvate m+3 exceeds citrate m+3/pyruvate m+3, indicating that pyruvate enters the TCA cycle predominantly by PDH (Extended Data Fig. 4c, d). Both ratios increased between GD9.5 and GD12.5 in the embryos, but not in placenta, where labelling declined slightly. These data indicate that pyruvate oxidation is timed differently in the embryo and placenta, lagging in the embryo by a few days.

The increased contribution of glucose to the TCA cycle in the embryo may reflect the development of oxidative organs such as the liver, heart and brain. To assess gene-expression signatures relevant to mitochondrial function, we analysed polyA plus RNAseq data from the ENCODE portal[21] from each of these organs across GD10.5–GD13.5. This revealed increased electron transport chain (ETC)-related transcript abundance over this period, particularly in the heart (Extended Data Fig. 4e). By contrast, most ETC-related transcripts declined in placenta over midgestation (Extended Data Fig. 4g). We then performed [U-13C] glucose infusions and analysed labelling in the brain, heart and liver. On GD12.5, TCA cycle intermediates displayed uniformly high labelling in the brain and heart, with less labelling in the liver (Fig. 3c). The ratio of citrate to pyruvate labelling in the heart increased during midgestation,

corresponding to enhanced expression of ETC subunits (Extended Data Fig. 4e, f); this is notable because heart development requires increased oxygenation and reduced HIF1α expression[2,22,23]. We also infused [U-13C] glutamine to assess metabolism of an alternative fuel. In contrast to [U-13C]glucose, infusion with [U-13C]glutamine resulted in higher labelling in metabolites from the GD12.5 liver compared with brain or heart (Fig. 3d). Kinetic experiments revealed consistent glutamine labelling in each organ, but higher labelling in glutamate in the liver throughout the time course (Extended Data Fig. 4h). These data indicate distinct patterns of fuel metabolism in developing embryonic organs.

## LIPT1 enables developmental metabolism

To test the importance of enhanced oxidative metabolism during midgestation, we examined the impact of lipoyltransferase-1 (LIPT1) deficiency in utero. LIPT1 transfers the essential lipoic acid cofactor onto mitochondrial 2-ketoacid dehydrogenases related to the TCA cycle, including PDH, α-ketoglutarate dehydrogenase (AKGDH), branched-chain ketoacid dehydrogenase (BCKDH) and 2-oxoadipate dehydrogenase[4,5]. We reported a patient with compound heterozygosity for pathogenic LIPT1 variants (N44S and S292X) and a phenotype of neurodevelopmental disability and seizures[5]. Mice homozygous for the N44S variant are detected at close to the expected Mendelian ratio at GD10.5 but absent by GD11.5, indicating embryonic lethality occurs between these days[5]. *Lipt1*[WT/N44S] mice are healthy, born at the expected frequency[5] and have similar metabolomic signatures to *Lipt1*[WT/WT] embryos at GD10.5 (Fig. 4a, Extended Data Fig. 5a), so we grouped *Lipt1*[WT/WT] and *Lipt1*[WT/N44S] together as 'healthy' in statistical analyses. *Lipt1*[N44S/N44S] conceptuses are viable but small on GD10.5 (Extended Data Fig. 5b). On GD10.5, *Lipt1*[N44S/N44S] conceptuses had metabolomic patterns consistent with deficiencies in lipoylation and the TCA cycle. A defect in AKGDH was apparent from accumulation of α-ketoglutarate in the placenta and embryo; depletion of products downstream of AKGDH also occurred in the embryos (Fig. 4a, Extended Data Fig. 5c).

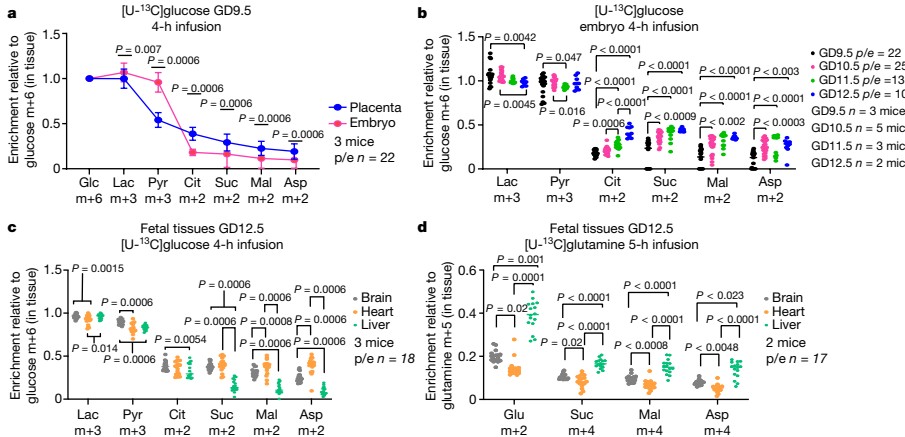

**Fig. 3 | Evolving labelling during midgestation. a**, Enrichments normalized to glucose m+6. **b**, Labelling from [U-¹³C]glucose between GD9.5 and GD12.5. **c**, **d**, Organ-specific enrichments at GD12.5. Statistical tests: paired $t$-tests followed by Holm–Sidak's multiple-comparisons adjustment (**a**); Kruskal-Wallis test followed by Dunn's multiple-comparisons adjustment or one-way ANOVA followed by Tukey's multiple-comparisons adjustment (**b**); linear mixed-effects analysis followed by Holm-Sidak's multiple-comparisons adjustment (**c**, between-tissue comparisons); or Welch's one-way ANOVA followed by the Dunnett's T3 multiple-comparisons adjustment or Kruskal–Wallis test followed by the Dunn's multiple comparisons adjustment (**d**). Data are mean ± s.d. Statistical tests were two-sided. Asp, aspartate; Cit, citrate; Glc, glucose; Lac, lactate; Mal, malate; Pyr, pyruvate; Suc, succinate.

Other abnormalities related to 2-ketoacid dehydrogenase dysfunction included accumulation of lysine and branched-chain ketoacids, particularly in the embryos (Fig. 4a Extended Data Fig. 5c).

We next performed infusions in pregnant dams at GD9.5 and GD10.5, first evaluating the capacity of *Lipt1*^N44S/N44S^ placentas to take up and transfer nutrients to the embryo. *Lipt1*^N44S/N44S^ placentas had no defects in taking up [U-¹³C]glucose or [U-¹³C]glutamine from the maternal circulation or transferring the label to the embryos (Extended Data Fig. 5d, e). Placental differentiation markers were largely conserved between healthy and Lipt1^N44S/N44S^ placentas (Extended Data Fig. 5f). From this, we conclude

that although LIPT1 deficiency alters placental metabolism, placental dysfunction is not the primary cause of lethality in the *Lipt1*^N44S/N44S^ embryos.

We also investigated the effects of LIPT1 deficiency on TCA cycle labelling at GD9.5 and GD10.5 (Fig. 4b, c, Extended Data Fig. 5g), just before the point of demise. *Lipt1*^N44S/N44S^ embryos were metabolically active and indistinguishable from healthy embryos in pyruvate or lactate labelling from ¹³C-glucose (Fig. 4b). However, TCA cycle labelling was suppressed in *Lipt1*^N44S/N44S^ tissues, particularly downstream of AKGDH, at both GD9.5 and GD10.5 (Fig. 4b, c, Extended Data Fig. 5g). Thus, *Lipt1*^N44S/N44S^ embryos do not induce TCA cycle labelling just prior to their midgestation demise.

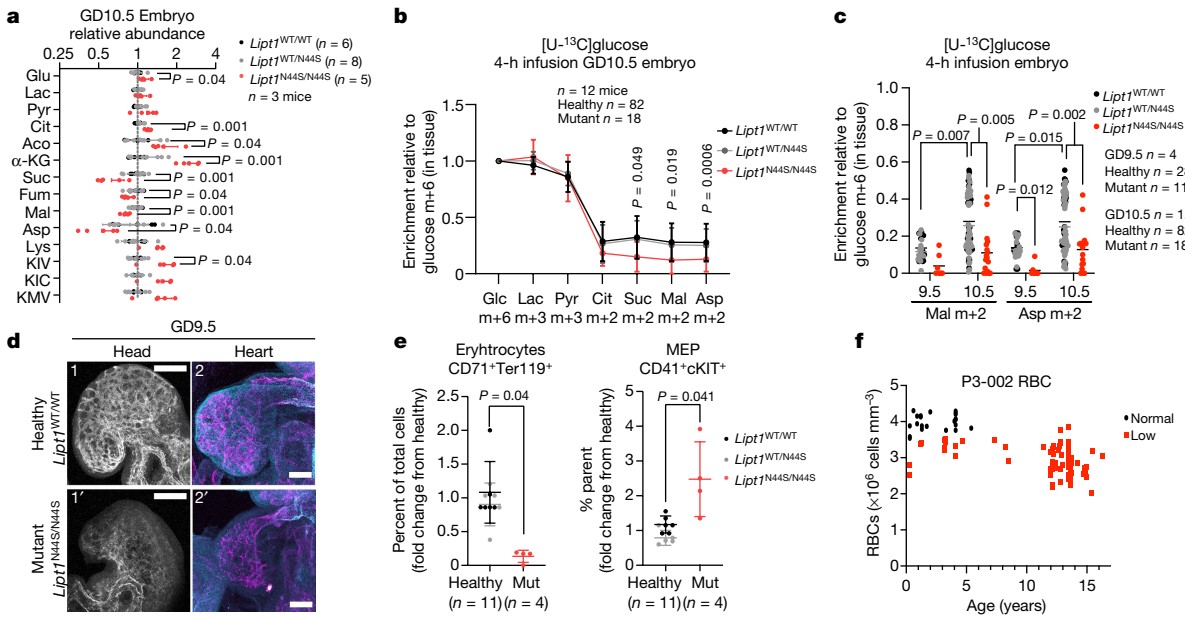

**Fig. 4 | *Lipt1* deficiency impairs embryo metabolism, growth and erythropoiesis. a**, Relevant metabolites in embryos of the indicated genotypes. **b**, **c**, Labelling from [U-¹³C]glucose. **d**, Endothelial cells in the head stained for PECAM1 and endomucin (1 and 1′; scale bar, 200 μm), and endothelial cells in the heart stained for PECAM1 and endomucin (magenta) and connexin 40 (cyan) (2 and 2′; scale bar, 100 μm). Images are representative of $n$ = 3 dams, and the following numbers of embryos: *Lipt1*^WT/WT^ ($n$ = 5), *Lipt1*^WT/N44S^ ($n$ = 4), *Lipt1*^N44S/N44S^ (n = 8). **e**, Quantification of cells from dissociated GD10.5 whole embryos stained with antibodies against CD71, TER119, CD41 and c-Kit. Erythrocytes express CD71 and

TER119, and myeloid–erythroid progenitors (MEP) express CD41 and c-Kit. '% parent' indicates the proportion of CD71⁻ TER119⁻ cells that stained CD41⁺ c-Kit⁺. **f**, Longitudinal red blood cell (RBC) measurements in a LIPT1-deficient patient. Statistical tests: Student's $t$-tests (**a**); log₂-transformation followed by Holm-Sidak's multiple-comparisons adjustment (**b**); Mann–Whitney tests followed by Holm-Sidak's multiple-comparisons adjustment (**e**); Kruskal–Wallis tests followed by the Dunn's multiple-comparisons adjustment (**c**). Data are ± s.d. Statistical tests were two-sided. α-KG, α-ketoglutarate; Aco, aconitase; Fum, fumarate; Lys, lysine; KIV, a-ketoisovalerate; KIC, α-ketoisocaproate; KMV, α-keto-β-methylvalerate.

Finally, we assessed development in these embryos. Somite counts were indistinguishable among the genotypes at GD9.5 (Extended Data Fig. 6a). The initial formation and patterning of blood vessels was normal, and blood vessels were present throughout the *Lipt1*[N44S/N44S] embryos (Extended Data Fig. 6b). Vessel maturation as assessed by the flow-responsive marker Connexin 40, was also normal (Extended Data Fig. 6b). However, both the brain and heart were smaller in the mutants (Fig. 4d, Extended Data Fig. 6b). We also assessed erythropoiesis by performing flow cytometry on cells from dissociated embryos using cell surface markers. We observed a decreased abundance of CD71[+]TER119[+] fetal erythrocytes and an increased abundance of CD41[+]c-Kit[+] myeloid–erythroid progenitors in *LIPT1*[N44S/N44S] embryos, suggesting impaired erythrocyte differentiation (Fig. 4e, Extended Data Figs. 6c, d, 7). To examine the human relevance of this observation, we reviewed 15 years of clinical records from our LIPT1-deficient patient, and found that she suffers from chonic, unexplained anaemia (Fig. 4f) despite normal iron, folate and vitamin B12 levels. Her platelet and white blood cell counts are preserved (Extended Data Fig. 6e, f), suggesting a particular defect in the erythroid lineage.

## Conclusions

Metabolic defects and exposure to metabolic inhibitors[16] can result in human congenital anomalies, emphasizing the importance of precise metabolic control during fetal development. Although resources exist to assess gene-expression and epigenetic signatures throughout development[24,25], understanding the developmental consequences of metabolic defects will benefit from methods to assess metabolism directly in utero. In this Article, we report metabolic features that evolve during midgestation in placenta and embryo, with both tissues undergoing extensive but largely distinct changes. The metabolic differences are consistent with requirements for rapid growth, dramatically divergent cellular composition of these tissues, and evolving celluar environments. Compartment-specific labelling differences in G6P and other metabolites indicate localized placental carbohydrate metabolism that may have little direct effect on embryonic glucose metabolism and possible differences in how each compartment meets its growth requirements. In the embryo, glucose supplies glycolysis, the pentose phosphate pathway and an expanding purine pool, all of which are rapidly labelled from glucose in the maternal circulation.

The contribution of maternally derived nutrients to the embryonic TCA cycle increases as midgestation progresses beyond GD9.5[26]. We thus sought to examine the metabolic effects of a human genomic variant that interrupts this process. LIPT1 activates multiple enzymes responsible for providing respiratory substrates to the TCA cycle, and human LIPT1 deficiency results in developmental anomalies in oxidative organs including the brain. In mice, we find that LIPT1 is required for precisely timed changes in mitochondrial metabolism necessary for development past GD10.5; LIPT1 mutants persist for about one day after TCA cycle labelling increases in wild-type counterparts, and then die. Embryonic demise involves delayed or defective development in tissues such as the heart that have enhanced pyruvate oxidation over this gestational time frame, and erythrocytes, whose development requires mitochondrial function[27]. Of note, the metabolic fate of pyruvate has been suggested to inform development in some contexts, with persistent conversion to lactate associated with stem cell expansion and oxidation in the TCA cycle associated with differentiation[28–30]. Observing metabolic pathways at the level of individual embryonic organs should provide an efficient approach to identify pathways that support spatiotemporal developmental programmes.

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

## Methods

### Materials

Materials were obtained as follows: [U-$^{13}$C]glucose (Cambridge Isotopes, CLM-1396), [U-$^{13}$C]glutamine (Cambridge Isotopes, CLM-1822), C57BL/6J (UTSW Mouse Breeding Core or Jackson Labs) and *Lipt1*$^{N44S}$ knock-in mice (developed in-house)[5].

### Subject information and clinical data

The LIPT1-deficient individual who provided clinical data in Fig. 4f, Extended Data Fig. 6e, f was described previously[5]. This patient was enrolled in a prospective, non-randomized, non-blinded observational study whose overarching goal is to discover new metabolic disease-associated genes in patients of any age, and to characterize the metabolic phenotype in these patients (NCT02650622). The study was approved by the Institutional Review Board (IRB) at University of Texas Southwestern Medical Center (UTSW), and written informed consent was obtained from the patient's parents. Patients and family members eligible for the study are identified at UTSW, its affiliated hospitals, and other collaborating hospitals. After enrollment, study subjects provide blood for metabolomics and genomics, and a research-based integrated analysis of the data allows potentially pathogenic genomic variants to be prioritized for functional analysis in the laboratory. The study is purely observational in that no therapeutic interventions are proposed, although patients are followed longitudinally to understand each disease's natural history and the effects of therapies instituted as a part of routine clinical care. A total enrollment of over 1,500 patients is planned with the intention of representing many rare conditions within the cohort.

### Reference datasets and data processing

Data for fetal tissues during midgestation are available from the ENCODE[21,35,40] project Mouse Development Matrix (https://www.encodeproject.org/mouse-development-matrix). We downloaded the tsv files from the polyA plus RNAseq assay with the following identifiers: ENCFF262TPS (E11.5 liver -1), ENCFF414APX (E11.5 liver-2), ENCFF173NFQ (E12.5 liver-1), ENCFF144DHB (E12.5 liver-2), ENCFF971KKK (E13.5 liver-1), ENCFF042DVY (E13.5 liver-2), ENCFF-770SOB (E10.5 heart-1), ENCFF351QKG (E10.5 heart-2), ENCFF159DWP (E11.5 heart-1), ENCFF168UJM (E11.5 heart-2), ENCFF484QWQ (E12.5 heart-1), ENCFF329HOZ (E12.5 heart-2), ENCFF148BEQ (E13.5 heart-1), ENCFF836QQS (E13.5 heart-2), ENCFF145PTV (E10.5 forebrain-1), ENCFF476ADM (E10.5 forebrain-2), ENCFF606UHO (E11.5 forebrain-1), ENCFF434CSI (E11.5 forebrain-2), ENCFF928MQD (E12.5 forebrain-1), ENCFF046RSQ (E12.5 forebrain-2), ENCFF960KJV (E13.5 forebrain-1), ENCFF356CTG (E13.5 forebrain-2). Placenta RNA transcript abundance was obtained from Gene Expression Omnibus (GEO) accession code GSE100053. Expression data were filtered based on known metabolic genes[37-39] and human–mouse gene mapping was based on the HomoloGene database (https://www.ncbi.nlm.nih.gov/homologene).

Placental gene-expression data were obtained from the GEO repository (https://www.ncbi.nlm.nih.gov/gds) using the GEOquery package[36] (https://doi.org/10.18129/B9.bioc.GEOquery) v2.62.1 from BioConductor release (3.14) (https://www.bioconductor.org/). Data were filtered based on known metabolic genes[37-39] and sorted by Kyoto Encyclopedia of Genes and Genomes pathway annotation in the metaboAnalyst_KEGG R package (https://github.com/xia-lab/MetaboAnalystR). Human–mouse gene mapping was based on the HomoloGene database (https://www.ncbi.nlm.nih.gov/homologene).

### Animal studies

All procedures were approved by the UT Southwestern Animal Care and Use Committee (IACUC) in accordance with *The Guide for the Care and Use of Laboratory Animals*. All mice were housed in a pathogen free environment (temperature 20–26 °C, humidity 30–70%) with a 12 h:12 h light:dark cycle and fed chow diet (Teklad 2916) ad libitum. Healthy 8–15 week old, naïve pregnant females were set up for mating between 05:00 and 07:00 with proven studs of the appropriate genotype. The following morning, females displaying vaginal plugs were identified as pregnant and moved to a new cage until the indicated gestational day.

### Metabolomic analysis

All sample collection took place between 09:00 and 11:00 with no prior fasting of the pregnant dams. Mice were initially anaesthetized using isoflurane and samples were dissected in cold sodium chloride irrigating solution (Baxter) and snap frozen in liquid nitrogen. Whole embryos and placentas were homogenized manually with a rubber dounce homogenizer in ice-cold acetonitrile:water (80:20). Samples were flash frozen 3 times in liquid nitrogen and then centrifuged at 16,000*g* for 10 min at 4 °C. Supernatants were subject to BCA analysis and normalized to 70 μg ml$^{-1}$ and placed in LC–MS vials. Metabolite analysis used a Vanquish UHPLC coupled to a Thermo Scientific QExactive HF-X hybrid quadrupole orbitrap high-resolution mass spectrometer (HRMS) as performed previously[31]. Pooled samples were generated from an equal mixture of all individual samples and analysed using individual positive- and negative-polarity spectrometry ddHRMS/MS acquisition methods for high-confidence metabolite ID. Metabolite identities were confirmed in three ways: (1) precursor ion *m/z* was matched within 5 ppm of theoretical mass predicted by the chemical formula; (2) fragment ion spectra were matched within a 5 ppm tolerance to known metabolite fragments; and (3) the retention time of metabolites was within 5% of the retention time of a purified standard run with the same chromatographic method. LC-MS/MS data were collected using SCIEX Analyst v1.6.3 and Thermo Scientific XCalibur 4.1.50 and data analysed using SCIEX Multiquant v2.1.1, and Thermo Scientific Trace Finder v5.1. Relative metabolite abundance was determined by integrating the chromatographic peak area of the precursor ion searched within a 5 ppm tolerance and then normalized to total ion count (TIC). Statistical analysis for generation of PCA plots, heatmaps, differential abundances and MSOA were performed using MetaboAnalyst 5.0 (https://www.metaboanalyst.ca). Data were log-transformed and auto-scaled prior to the analysis. Additional heatmaps (Fig. 1e, Extended Data Fig. 2e) were generated using GraphPad Prism 9.0.1. For $^{13}$C studies, observed distributions of mass isotopologues were corrected for natural isotope abundances using a customized R script, which can be found at the GitHub repository (https://github.com/wencgu/nac). The script was written by adapting the AccuCor algorithm v0.2.4[32].

### Pregnant mouse infusions

All infusions took place between 09:00 and 11:00 with no prior fasting of the pregnant dams. Mice were initially anaesthetized using ketamine and xylazine (120 mg kg$^{-1}$ and 16 mg kg$^{-1}$, respectively, intraperitoneally) and maintained under anaesthesia using subsequent doses of ketamine (20 mg kg$^{-1}$, intraperitoneally) as needed. Catheters (25-gauge) were inserted into the tail vein and isotope infusions began immediately after retro-orbital blood draw to mark time zero. In the glucose infusions, the total dose was 2.48 g kg$^{-1}$ dissolved in 750 μl normal saline and administered with a bolus of 62.5 μl min$^{-1}$ for 1 min followed by an infusion rate of 2.5 μl min$^{-1}$ for 3–4 h. Retro-orbital blood draws were taken throughout the infusion to monitor tracer enrichment in maternal blood. Glutamine infusions used a total dose of 1.73 g kg$^{-1}$ dissolved in 1,500 μl normal saline administered as a bolus of 147 μl min$^{-1}$ for 1 min followed by an infusion rate of 3 μl min$^{-1}$ for 5 h. Mice were euthanized at the end of the infusion, then the uterus was removed and placentas and embryos dissected in cold sodium chloride irrigating solution and frozen in liquid nitrogen. Care was taken during infusions not to increase nutrient concentrations over pre-infusion levels.

## Serial caesarian-section surgery

For serial caesarian sections, the infusion parameters were the same as described above with the following alterations: (1) Serial caesarian-section infusions did not include a bolus; (2) the infusion rate was increased to 5 µl min$^{-1}$ in order to obtain sufficient labelling. Although the patterns of data for serial caesarian-sections matched what we observed in the 4 h infusions, the overall labelling was somewhat lower and for this reason we did not compare serial caesarian-section data to data from longer infusions. After cannulation of the tail vein and retro-orbital blood draw for time zero, the lower abdomen of the pregnant dam was opened with a small incision. The uterus was removed from the peritoneal cavity and the conceptus nearest to one of the ovaries was dissected away from the uterus and further dissected into placenta and embryo in cold sodium chloride irrigating solution and then frozen in liquid nitrogen. The peritoneal cavity was flushed with sodium chloride irrigating solution, covered with gauze, and periodically rinsed with irrigating solution throughout the remainder of the surgery. The infusion was initiated and a single conceptus was dissected in a similar manner at the indicated time points until all embryos had been dissected or the 3 h time point was reached.

## Gas chromatography mass spectrometry (GCMS)

Gas chromatography–mass spectrometry (GCMS) was used to identify glucose, pyruvate, lactate, citrate, succinate, malate and aspartate. These metabolites were also identified using liquid chromatography–mass spectrometry (LC–MS) and enrichment values were similar. Blood samples obtained during the infusion were chilled on ice for 5–10 min and then flash frozen in liquid nitrogen. Aliquots of 10–20 µl were added to 80:20 acetonitrile:water for extraction. Frozen tissues (whole embryo and whole placenta) were added to 80:20 acetonitrile:water and extracted to analyse $^{13}$C enrichment. Samples were manually disrupted using a rubber dounce homogenizer, subjected to three freeze–thaw cycles, then centrifuged at 16,000$g$ for 15 min to precipitate macromolecules. For GCMS, 1 µl D$_{27}$-myristic acid was added as an internal control, supernatants were evaporated, then re-suspended in 30 µl anhydrous pyridine with 10 mg ml$^{-1}$ methoxyamine and incubated at room temperature overnight. The following morning, the samples were incubated at 70 °C for 10–15 min and then centrifuged at 16,000$g$ for 10 min. The supernatant was transferred to a pre-prepared GC/MS autoinjector vial containing 70 µl $N$-($tert$-butyldimethylsilyl)-$N$-methyltrifluoroacetamide (MTBSTFA) derivatization reagent. The samples were incubated at 70 °C for 1 h after which aliquots of 1 µl were injected for analysis. Samples were analysed using either an Agilent 6890 or 7890 gas chromatograph coupled to an Agilent 5973N or 5975C Mass Selective Detector, respectively. GC–MS data were collected and analysed using Agilent ChemStation E02.02.1431. The observed distributions of mass isotopologues were corrected for natural isotope abundances using a customized R script, which can be found at the GitHub repository (https://github.com/wencgu/nac). The script was written by adapting the AccuCor algorithm v0.2.4[32].

## Gene expression

Total RNA was extracted from placental tissue using TRIzol Reagent (Thermo Fisher Scientific cat. no. 15596026). RNA (3,250 ng) was used as a template for a 70 µl cDNA synthesis reaction using TaqMan Reverse Transcription Reagents (Thermo Scientific cat. no. N8080234) according to the manufacturer's instructions. cDNA was diluted 1:1 in nuclease-free water and plated at a final volume of 4 µl in a 384-well plate. Primers for placental markers were as described[33] and diluted to a final concentration of 2.5 µM. Primers were mixed with iTaq Universal SYBR Supermix (Bio-Rad cat. no. 1725121) and plated at a volume of 6 µl for a total reaction volume of 10 µl. Plates were run in a Bio-Rad CFX384 Touch Real-Time PCR Detection machine using the following protocol: (1) polymerase activation: 95 °C hold for 30 min; (2) PCR phase, 40 cycles: 95 °C hold for 5 s, 60 °C hold for 30 s; (3) melt curve, instrument default settings. Relative fold induction was computed using the $\Delta\Delta C_T$ method, as described[34].

Embryo RNA sequencing data were downloaded from the ENCODE Mouse Development Matrix[35] (https://www.encodeproject.org/). PolyA plus RNA-seq data were obtained for fetal heart, forebrain and liver from GD10.5-GD12.5 (not all days are available for liver). Placenta RNA transcript abundance was obtained from GEO accession code GSE100053 using the GEOquery package[36] (https://doi.org/10.18129/B9.bioc.GEOquery) v2.62.1 from BioConductor release (3.14) (https://www.bioconductor.org/). Data were filtered based on known metabolic genes[37–39] and sorted by Kyoto Encyclopedia of Genes and Genomes pathway annotation in the metaboAnalyst_KEGG R package (https://github.com/xia-lab/MetaboAnalystR). Human–mouse gene mapping was based on the HomoloGene database (https://www.ncbi.nlm.nih.gov/homologene).

## Flow cytometry

Whole embryos were collected from GD10.5 pregnant mice into 1× PBS and mechanically disrupted using disposable pestles (VWR) and then filtered through a 40-µM cell strainer to remove clumps. Antibody staining was performed for 20 min on ice, followed by washing with HBSS (Invitrogen) and centrifugation at 200$g$ for 5 min. Cells were stained with directly conjugated antibodies against mouse CD71 (FITC-R17.217.1.4 Biolegend, 1:100), mouse Ter119 (APC-TER-119 TONBO, 1:100), mouse CD41 (PE/Cy7-MWReg30 Biolegend, 1:100) and mouse CD117 (cKIT-APC-eFlour 780-Invitrogen, 1:100). All cells were gated for forward and side scatter and gated for live cells based on DAPI (1 µg ml$^{-1}$; Sigma, eFlour-450A). Erythrocytes were cells that were negative for CD117 (c-KIT), and positive for CD71 and Ter119. Myeloid–erythroid progenitors were negative for CD71 and TER119 and positive for CD41 and CD117 (c-KIT). Cells were examined on an LSRFortessa cell analyser (Becton Dickinson) and figures were generated using BD FACSDiva 8.0 and FlowJo v10.

## Whole-mount immunofluorescent staining

Pregnant females at the desired developmental stage were euthanized by carbon dioxide asphyxiation and the uterus and extra-embryonic tissues were removed. Yolk sacs were used for genotyping and somites were counted. Embryos were fixed in 4% paraformaldehyde for 1 h at 25 °C or 4 °C overnight. Fixed embryos were washed at least 3 times with 1× PBS and dehydrated through a series of methanol or ethanol (25%, 50%, 75% and 100%, two times), permeabilized using 1% Triton X-100 (Fisher Bioreagents, cat. no. BP151-100) in PBS for 1.5–2 h at 25 °C, then blocked using CAS Block (Life Technologies, cat. no. 008120) for 2 h. Embryos were incubated in primary antibodies diluted in CAS Block overnight at 4 °C: Rat-anti-PECAM1 (1:100, BD, Biosciences, cat. no. 553370), Rat-anti-endomucin (1:100, Santa Cruz, sc-65495) and Rabbit-anti-connexin 40 (1:100, Alpha Diagnostics International, cat. no. CX-40A). Embryos were washed with 1× PBS then incubated with secondary antibodies diluted in CAS Block at 1:250 overnight at 4 °C: donkey-anti-rat 488 (Invitrogen, cat. no. A21208), donkey-anti-rabbit 555 (Invitrogen, cat. no. A31572). Embryos were washed in 1× PBS, then dehydrated to 100% methanol through a methanol series (25%, 50%, 75%, 100% two times, 10 min each), cleared in a 1:2 benzyl alcohol:benzyl benzoate (BABB) solution, and mounted in BABB in 5 mm Thick Microscopy slides (Chang Biosciences, Rb167104D_1) and cover slipped. Images were obtained using a LSM700 Ziess confocal microscope with the Carl Zeiss ZEN 2011 software. If images of the dissected heart were desired, whole embryos were rehydrated through a methanol series into PBS, hearts were dissected and placed in a 1.5 mm 2-well concavity slide (Electron Microscopy Sciences, cat. no. 71878-03) containing PBS. Whole-heart images were obtained using a Ziess Images M2 with an Axiocam 506 mono camera attached with the Carl Zeiss ZEN 2011 software. For sectioned samples, paraffin embedded samples were transverse sectioned at 5 µm and stained with haematoxylin and eosin.

## Statistical analysis

During flow cytometry, isotope tracing, metabolomics, quantitative PCR, tissue weights, somite counts and histology experiments, the data were analysed in a manner blinded to sample genotype. A.S. collected the samples and then passed them to A. Tasdogan. for flow cytometry, or to I.M.-M. and M.A.C. for histology and immunofluorescence, and A. Tarangelo. for quantitative PCR. A.S. processed samples for mass spectrometry and analysed data. After the patterns had been analysed in each of these experiments, D. Dumesnil. provided the genotype information so results could be interpreted. For experiments in wild-type mice, no blinding was performed on placentas versus embryos because A.S. performed these experiments and analysed the data. For gene-expression studies from publicly available datasets, no blinding was performed.

Mice were allocated to experiments randomly and samples were processed in an arbitrary order, but formal randomization techniques were not used. Samples sizes were not pre-determined based on statistical power calculations but were based on our experience with these assays. For most experiments, the minimum number of mice was 3, with some exceptions where the embryo/placenta numbers were $n \geq 10$. No data were excluded; however, sometimes the small sample size was below the threshold for metabolomic analysis. In those instances, data that could be obtained from maternal blood or other tissues were used. These samples were not used during direct comparisons of embryo relative to its own placenta if one of the samples was absent.

Prior to analysing the statistical significance of differences among groups, we tested whether data were normally distributed and whether variance was similar among groups. To test for normality, we performed the Shapiro–Wilk tests when $3 \leq n < 20$ or D'Agostino omnibus tests when $n \geq 20$. To test whether variability significantly differed among groups we performed $F$-tests (for experiments with two groups) or Levene's median tests (for experiments with more than two groups). When the data significantly deviated from normality or variability significantly differed among conditions, we $\log_2$-transformed the data and tested again for normality and variability. If the transformed data no longer significantly deviated from normality and equal variability, we performed parametric tests on the transformed data. If $\log_2$-transformation was not possible or the transformed data still significantly deviated from normality or equal variability, we performed non-parametric tests on the non-transformed data.

When data or $\log_2$-transformed data were normal and equally variable, statistical analyses were performed using Student's $t$-tests or paired $t$-tests (when there were two groups), one-way ANOVAs or repeated measures one-way ANOVAs (when there were more than two groups), two-way repeated measures ANOVAs (when there were two or more groups with multiple metabolites or time points), or mixed effects models (when there were missing values but the data otherwise met the assumptions for a one-way or two-way repeated measures ANOVA). When the data or $\log_2$-transformed data were normal but unequally variable, statistical analyses were performed using Welch's $t$-tests (when there were two groups) or Welch's one-way ANOVAs followed by the Dunnett's T3 tests for multiple-comparisons adjustment (when there were more than two groups). When the data and $\log_2$-transformed data were abnormal or unequally variable, statistical analysis was performed using Mann–Whitney or Wilcoxon matched pairs signed rank tests (when there were two groups) or Kruskal–Wallis tests (when there were more than two groups). $P$-values from multiple comparisons were adjusted using Tukey's (when there were more than two groups and all of the comparisons were of interest) or Sidak's method (when there were more than two groups and planned comparisons) after ANOVAs or mixed effects models, or Dunn's method after Kruskal–Wallis tests. Holm–Sidak's method was used to adjust comparisons involving multiple metabolites between two conditions. A linear regression or nonlinear curve fitting method, plateau followed by one-phase association, was used to fit the time series data and the extra sum-of-squares $F$-test was used to assess if there was difference between two fitted lines/curves. Multiple line/curve fitting $P$-values were adjusted using the Holm–Sidak method. Statistical tests were performed using GraphPad Prism V9.0.1 or R 4.0.2.

## Reporting summary

Further information on research design is available in the Nature Research Reporting Summary linked to this paper.

## Data availability

Source data are provided with this paper.

## Code availability

Mass isotopologues were corrected for natural isotope abundances using a customized R script, which can be found at the GitHub repository (https://github.com/wencgu/nac). The script was written by adapting the AccuCor algorithm v0.2.4[32].

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

**Acknowledgements** This manuscript is dedicated to Gerardo Guevara who was an excellent laboratory member and friend and whom we miss dearly. K. Dickerson helped interpret the patient's anaemia and A. B. Jaffe provided critical feedback. A.S. is a Ruth L. Kirschstein National Research Service Award Postdoctoral Fellow from the Eunice Kennedy Shriver National Institute of Child Health and Human Development (F32HD096786-01). B.F. is supported by a Career Transition Award from the National Cancer Institute (K99CA237724-01A1). R.J.D. is supported by the H.H.M.I. Investigator Program, N.C.I. Grant R35CA22044901, the Baldridge Family and the Robert L. Moody Sr Faculty Scholar endowment. A.R. is supported by the Victorian Cancer Agency Early Career Research Fellowship. S.K. is a a Ruth L. Kirschstein National Research Service Award Predoctoral Fellow (F30CA254150-01A1). T.J.R. (K00CA212230) and A. Tarangelo. (K00CA234650) are NCI Predoctoral to Postdoctoral Fellows. The schematic of the midgestation (Fig. 1a) and infusion (Fig. 2a) procedures were generated using BioRender (https://biorender.com/). We also acknowledge the ENCODE Consortium[35,40] and the ENCODE production laboratory(s) for generating the datasets used in this study[21].

**Author contributions** A.S., B.F. and R.J.D. conceived of the project, designed experiments and interpreted data. A.S., W.G., A.R., S.K., A. Tarangelo., A. Tasdogan, G.G., I.M.-M., M.A.C. and D. Dumesnil performed experiments. T.P.M., L.G.Z., M.S.M.-S. and D. Do, performed metabolomics and LCMS experiments. A.S., T.J.R., P.P. and L.C. performed metabolic tracing and gene-expression analysis. A.S., M.N. and C.P. generated and bred mice for LIPT1 experiments. Z.Z. performed statistical analyses. A.S. and R.J.D. wrote and edited the manuscript with help from B.F., A.P., O.C., S.J.M. and H.A.S.

**Competing interests** R.J.D. is an advisor for Agios Pharmaceuticals and Vida Ventures and a co-founder of Atavistik Bio. The other authors declare no competing interests.

**Additional information**
**Correspondence and requests for materials** should be addressed to Ralph J. DeBerardinis.

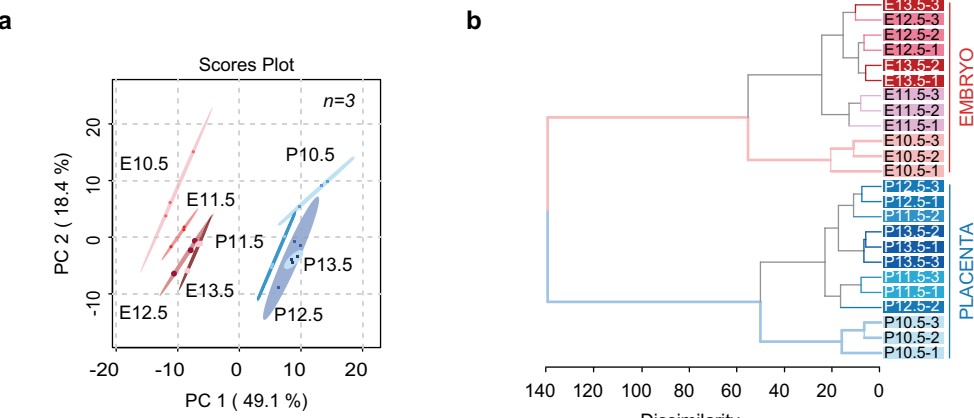

**Extended Data Fig. 1 | Dynamic metabolism in developmental tissues during midgestation.** (a) Principal component analysis and (b) Dendrogram analysis generated using Metaboanalyst 5.0 (https://www.metaboanalyst.ca/).

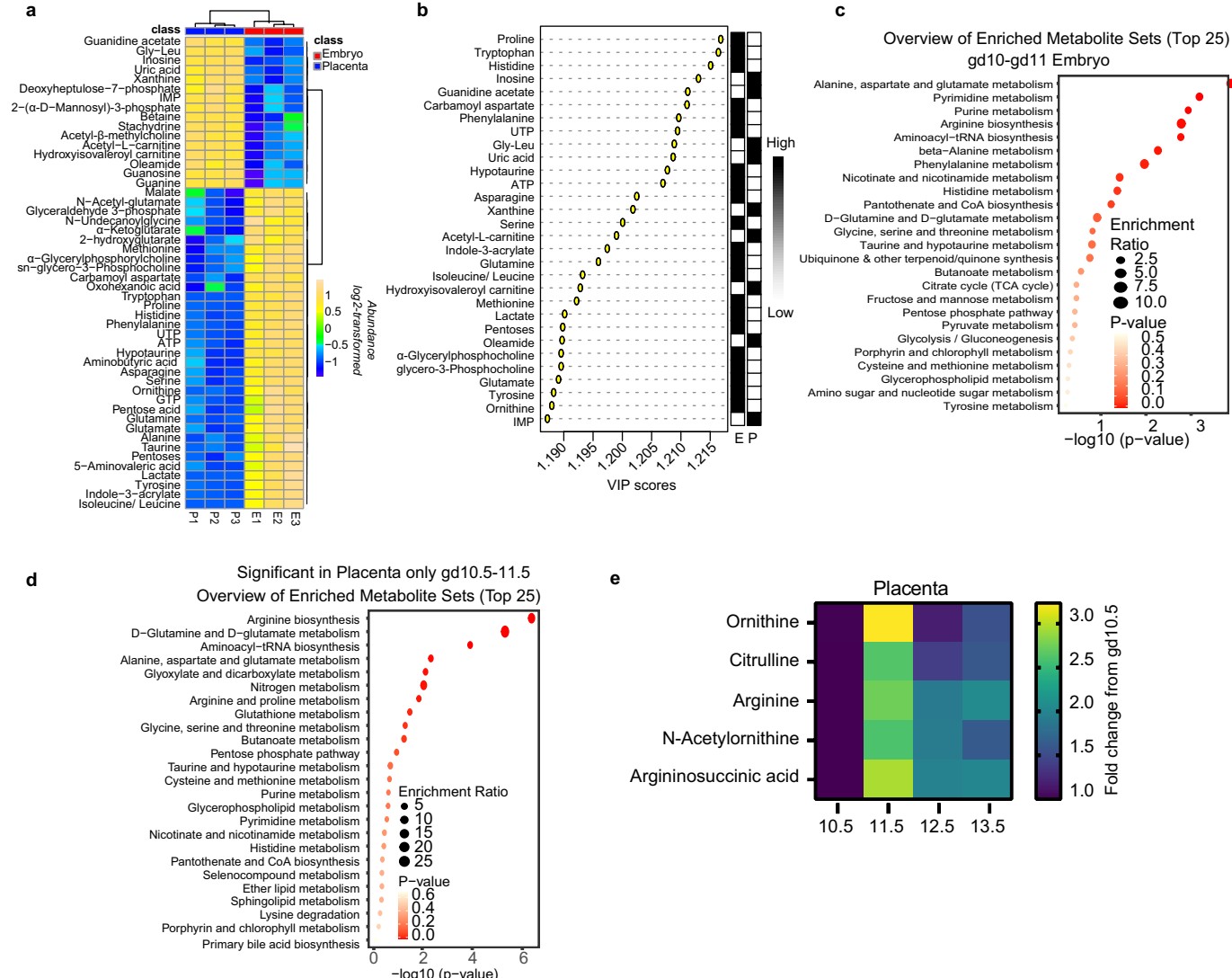

**Extended Data Fig. 2 | Tissue specific metabolic changes from gd10.5-gd11.5.** (a) Heatmap of top 50 metabolites significantly different between embryo and placenta. (b) Top 25 metabolites that contribute to the separation of gd10.5 embryo and placenta metabolic profiles. (c-d) Metabolic set overrepresentation analysis using metabolites in embryos (c) and placentas (d) differing in abundance (p < 0.05) between gd10.5-gd11.5. (e) Metabolites associated with the urea cycle transiently increase at gd11.5 and decrease the following two days. Statistical significance was determined using Student's t-tests. All data represent mean ± s.d. Statistical tests were two-sided.

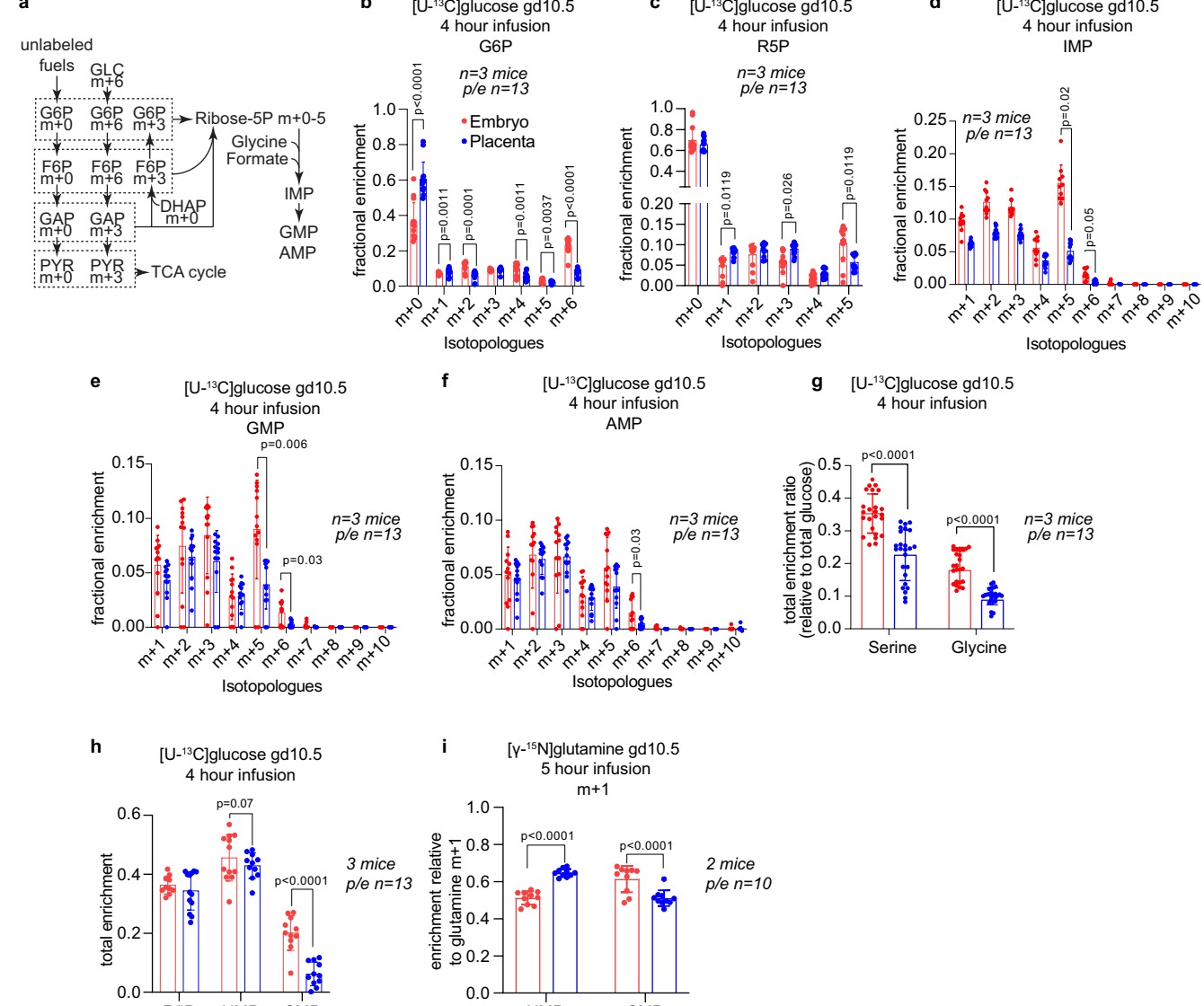

**Extended Data Fig. 3 | Glucose feeds purine synthesis in developmental tissues.** (a) Possible pathways contributing to labeling from [U-13C]glucose. (b-f) Isotopologues of G6P (b), R5P (c), IMP (d), GMP (e) and AMP (f) labeled with [U-13C]glucose. (g) Total enrichment (1-unlabeled fraction) of serine and glycine relative to total glucose enrichment. (h) Total enrichment in R5P (isobar with Ri5P/X5P), UMP and CMP labeled with [U-13C]glucose. (i) UMP and CMP m+1 enrichment from [γ-15N]glutamine. 15N-glutamine enrichment is normalized to glutamine m+1 to account for differences among compartments (see Fig. 2c). Statistical significance was determined using paired t-tests (b, c, h, and i) or Wilcoxon matched-pairs signed rank tests (b-g) followed by the Holm-Sidak's multiple comparisons adjustment (b-i). All data represent mean ± s.d. Statistical tests were two-sided.

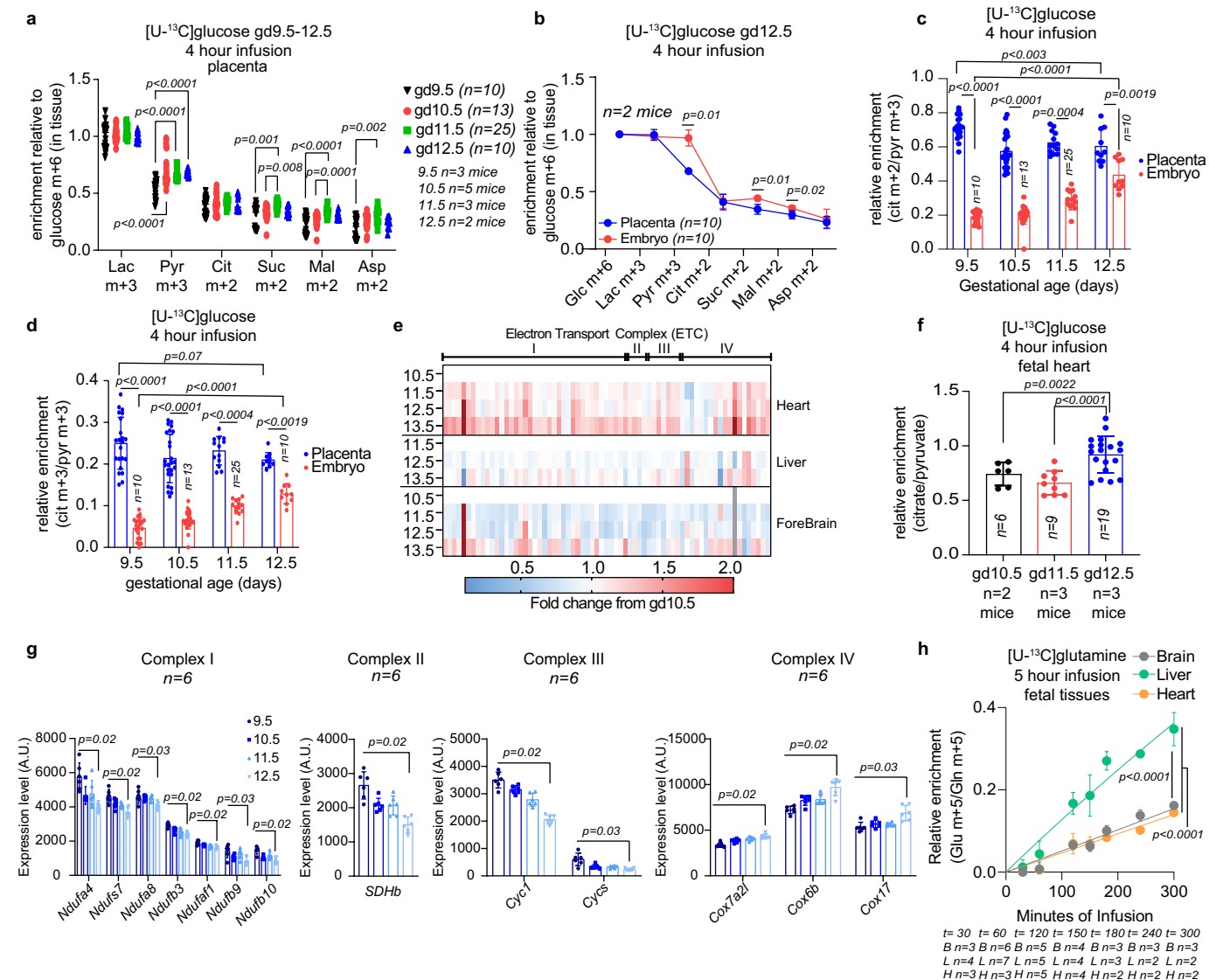

**Extended Data Fig. 4 | Tissue-specific TCA cycle metabolism in midgestation.** (a) Labeling from [U-¹³C]glucose between gd9.5-gd12.5 in placenta. (b) Enrichments normalized to glucose m+6 on gd12.5. (c-d) Daily citrate m+2/pyruvate m+3 (c) and citrate m+3/pyruvate m+3 (d) enrichment ratios. (e) ETC-related transcript counts (n = 2) in fetal tissues normalized to gd10.5 (gd11.5 in liver). (f) Daily total citrate/pyruvate enrichment ratio in fetal heart. (g) Placental ETC complex gene expression. (h) Enrichment ratio of glutamate m+5/glutamine m+5 in fetal tissues infused with [U-¹³C]glutamine.

Statistical significance was determined using Mann-Whitney tests (a, g), paired t-tests (b), or straight line least squares fitting (h) followed by the Holm-Sidak's multiple comparisons adjustment (a, b, g, and h), linear mixed-effects analysis (c,d) followed by the Sidak's (c-d; between-tissue comparisons) or Tukey's (c-d; between-time comparisons) multiple comparisons adjustment, or one-way ANOVA followed by the Tukey's multiple comparisons adjustment (f). All data represent mean ± s.d. Statistical tests were two-sided.

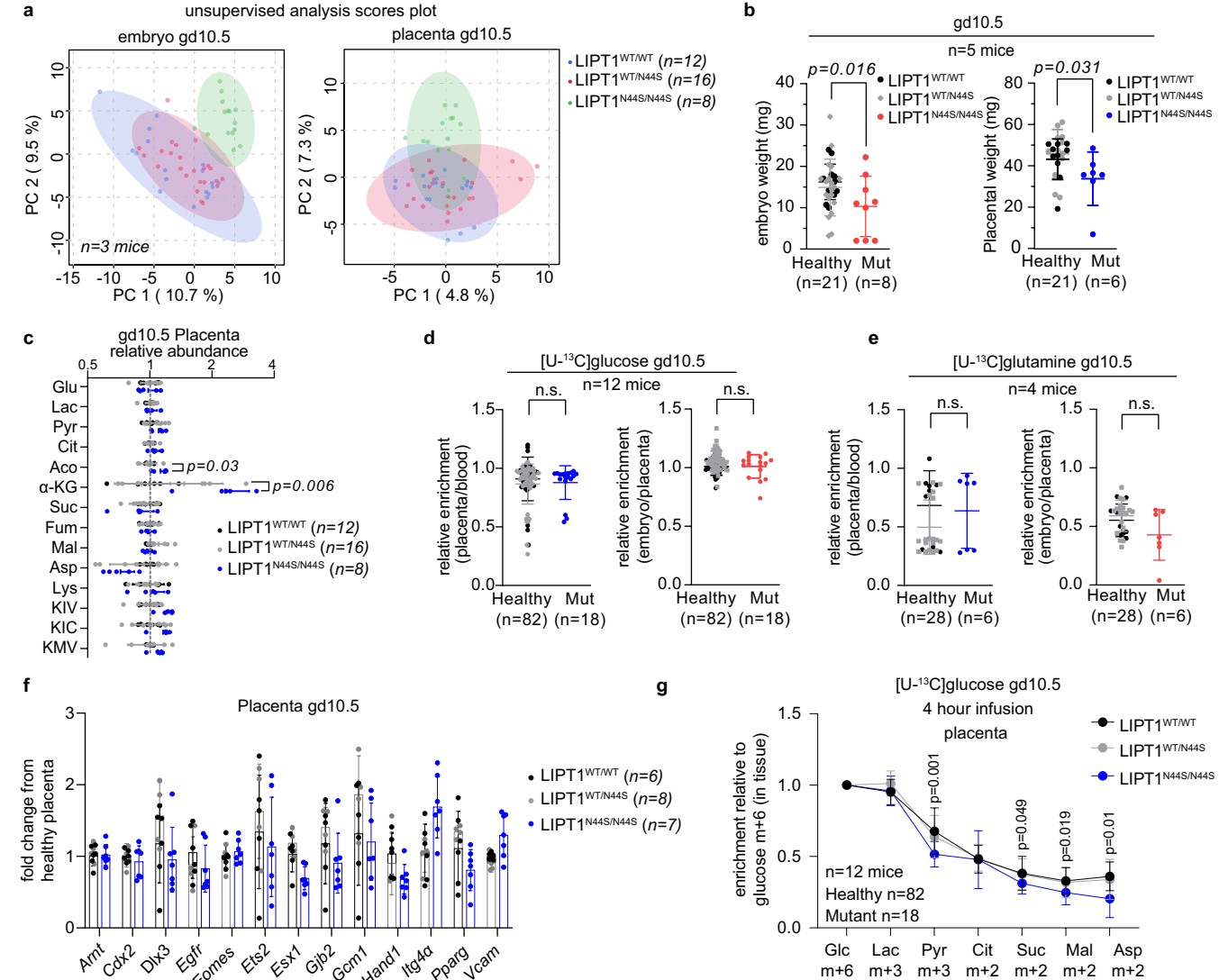

**Extended Data Fig. 5 | LIPT1 activity is critical for transition from gd10.5-gd11.5.** (a) PCA plots of metabolomics data and (b) tissue weights in litters arising from *Lipt1^{WT/N44S}* intercrosses. (c) Relevant metabolites in placentas of the indicated genotypes. (d-e) Placental uptake (left) and embryo transfer (right) of [U-¹³C]glucose (d) and [U-¹³C]glutamine (e). (f) Expression of placental markers. (g) Labeling from [U-¹³C]glucose in placentas of various *Lipt1* genotypes. Statistical significance was determined using two-way repeated measures ANOVA followed by the Sidak's multiple comparisons adjustment (b), Student's t-tests (c) or Mann-Whitney tests (d-g) followed by Holm-Sidak's multiple comparisons adjustment. All data represent mean ± s.d. Statistical tests were two-sided.

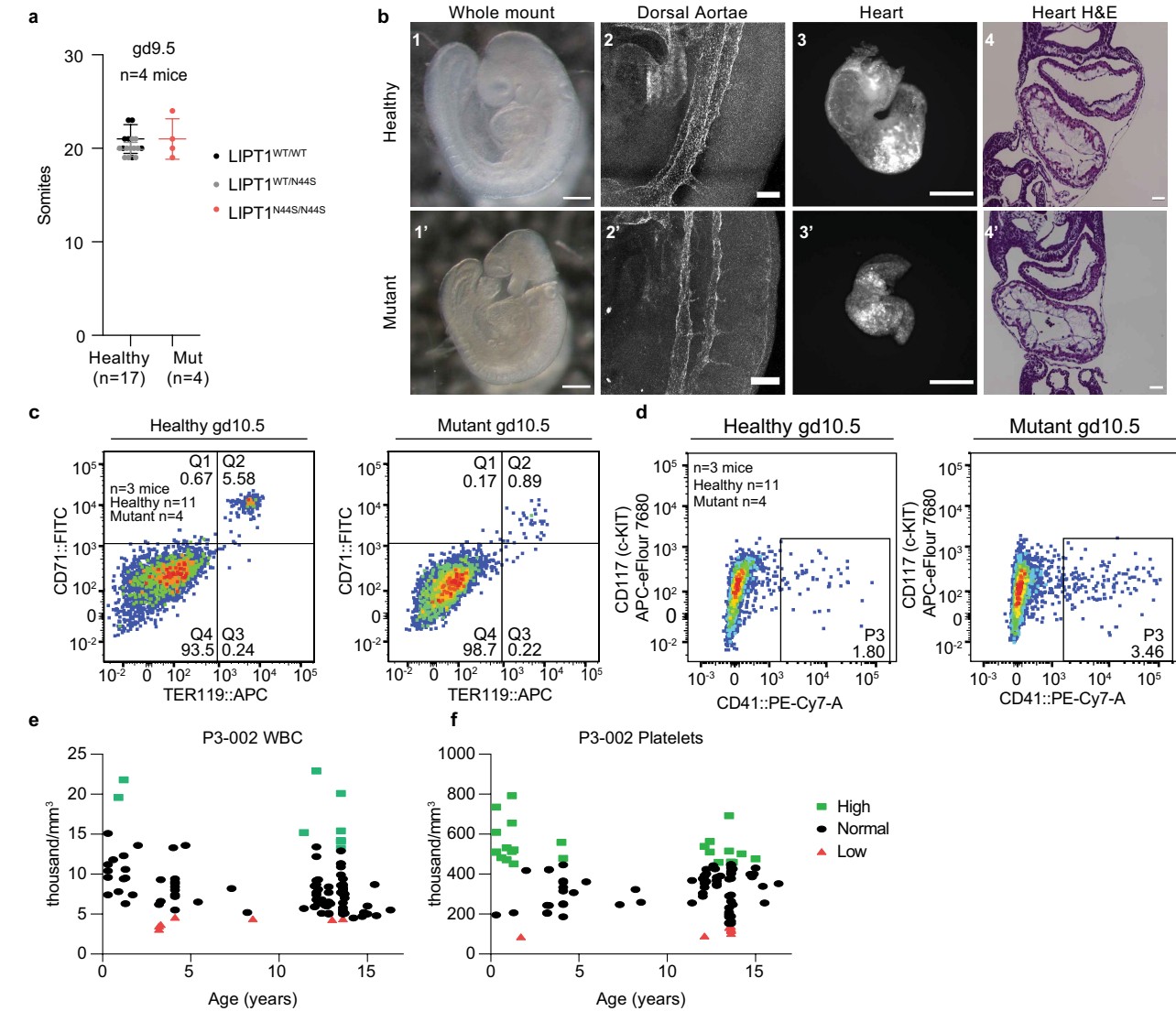

**Extended Data Fig. 6 | LIPT1 deficiency hinders organogenesis and erythropoiesis.** (a) Somite counts in embryos from litters arising from *Lipt1*<sup>WT/N44S</sup> intercrosses at gd9.5. (b) Brightfield whole mount images, scale bar = 500μm (1,1'); Dorsal Aortae stained with Connexin 40, scale bar = 100μm (2,2'); PE staining of whole hearts, scale bar = 300μm (3,3'); H&E staining of hearts, scale bar = 50μM (4,4'). All images from gd9.5 embryos. (c-d) Gd10.5 whole embryo cells stained with antibodies against the erythroid lineage markers CD71 and TER119 (c) and myeloid/erythroid progenitor markers, cKIT and CD41 (d). Flow cytometry was performed in 24 individual embryos (Healthy n = 11, Mutant n = 7) White blood cell (WBC) (e) and platelet (f) counts from a LIPT1-deficient patient. Statistical significance was determined using Student's t-tests followed by Holm-Sidak's multiple comparisons adjustment. All data represent mean ± s.d. Statistical tests were two-sided.

a

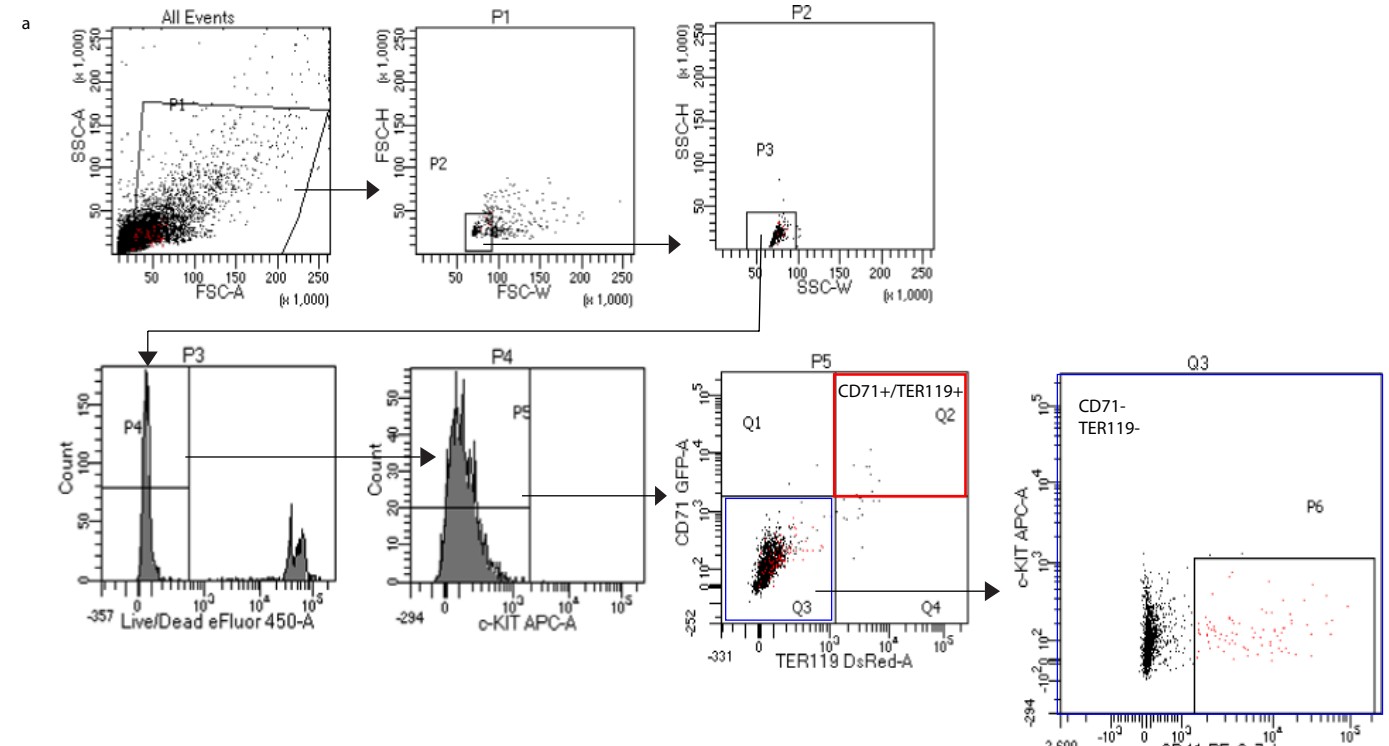

**Extended Data Fig. 7 | Flow cytometry gating strategies.** (a) Single cell suspensions from whole embryos were gated by forward and side scatter area (P1) then by forward scatter height and width (P2), then by side scatter height and width (P3). Cells had been stained with DAPI and live cells were gated as DAPI negative (P4), and then by CD117 (cKIT) negative (P5). Fetal erythrocytes were identified as CD71$^+$/TER119$^+$ (red box – P5:Q2), and myeloid/erythroid progenitors were gated as CD71$^-$/TER119$^-$ (blue box - P5:Q3) and also gated as cKIT$^+$/CD41$^+$ (P6).

# Reporting Summary

## Statistics

For all statistical analyses, confirm that the following items are present in the figure legend, table legend, main text, or Methods section.

| n/a | Confirmed | |
|---|---|---|
| ☐ | ☒ | The exact sample size ($n$) for each experimental group/condition, given as a discrete number and unit of measurement |
| ☐ | ☒ | A statement on whether measurements were taken from distinct samples or whether the same sample was measured repeatedly |
| ☐ | ☒ | The statistical test(s) used AND whether they are one- or two-sided *Only common tests should be described solely by name; describe more complex techniques in the Methods section.* |
| ☐ | ☒ | A description of all covariates tested |
| ☐ | ☒ | A description of any assumptions or corrections, such as tests of normality and adjustment for multiple comparisons |
| ☐ | ☒ | A full description of the statistical parameters including central tendency (e.g. means) or other basic estimates (e.g. regression coefficient) AND variation (e.g. standard deviation) or associated estimates of uncertainty (e.g. confidence intervals) |
| ☐ | ☒ | For null hypothesis testing, the test statistic (e.g. $F$, $t$, $r$) with confidence intervals, effect sizes, degrees of freedom and $P$ value noted *Give P values as exact values whenever suitable.* |
| ☒ | ☐ | For Bayesian analysis, information on the choice of priors and Markov chain Monte Carlo settings |
| ☒ | ☐ | For hierarchical and complex designs, identification of the appropriate level for tests and full reporting of outcomes |
| ☒ | ☐ | Estimates of effect sizes (e.g. Cohen's $d$, Pearson's $r$), indicating how they were calculated |

*Our web collection on statistics for biologists contains articles on many of the points above.*

## Software and code

Policy information about availability of computer code

| | |
|---|---|
| Data collection | Flow cytometry data were collected using LSRFortessa cell analyser (Becton Dickinson). GC-MS data were collected using Agilent ChemStation E02.02.1431, LC-MS/MS data were collected using SCIEX Analyst v1.6.3 and Thermo Scientific XCalibur 4.1.50. Quantitative PCR data were collected using a Bio-Rad CFX384 Touch Real-Time PCR Detection machine. Whole mount IF images were obtained using a LSM700 Ziess confocal microscope. Whole heart and H&E stained slides were imaged using a Ziess Images M2 with an Axiocam 506 mono camera attached. |
| Data analysis | GraphPad Prism V9.0.1 or R 4.0.2 with the stats, fBasics, car, and nparLD packages, Flow cytometry data analysis using BD FACSDiva 8.0, and FlowJo V10 (Treestar). GC-MS data analysis using Agilent ChemStation E02.02.1431. LC-MS/MS data analysis using SCIEX Multiquant v2.1.1 and Thermo Scientific Trace Finder 5.1, IF, whole heart and H&E images were analyzed with the Carl Zeiss ZEN 2011 software. Statistical analysis for generation of PCA plots, heatmaps, differential abundances and metabolic set overrepresentation analysis were performed using MetaboAnalyst 5.0 (https://www.metaboanalyst.ca). For 13C studies, observed distributions of mass isotopologues were corrected for natural isotope abundances using a customized R script, which can be found at the GitHub repository (https://github.com/wencgu/nac). The script was written by adapting the AccuCor algorithm v0.2.4. Gene expression data obtained using the GEOquery package v2.62. (DOI:10.18129/B9.bioc.GEOquery) from BioConductor release 3.14 (https://www.bioconductor.org/). Further data analysis was performed using the metaboAnalyst_KEGG R package (https://github.com/xia-lab/MetaboAnalystR) and the HomoloGene database ((https://www.ncbi.nlm.nih.gov/homologene). |

For manuscripts utilizing custom algorithms or software that are central to the research but not yet described in published literature, software must be made available to editors and reviewers. We strongly encourage code deposition in a community repository (e.g. GitHub). See the Nature Portfolio guidelines for submitting code & software for further information.

## Data

Policy information about availability of data

All manuscripts must include a data availability statement. This statement should provide the following information, where applicable:

- Accession codes, unique identifiers, or web links for publicly available datasets
- A description of any restrictions on data availability
- For clinical datasets or third party data, please ensure that the statement adheres to our policy

Data supporting the findings of this study are available within the article and its Supplementary Information files or from the corresponding author on request. Source Data for Figures 1-4 and Extended Data Figures 1-6 are provided with the paper as supplementary tables. RNAseq data for fetal tissues during midgestation are available from the ENCODE project Mouse Development Matrix (https://www.encodeproject.org/mouse-development-matrix). We downloaded the tsv files from the polyA plus RNAseq assay with the following identifiers: ENCFF262TPS (E11.5 liver -1), ENCFF414APX (E11.5 liver-2), ENCFF173NFQ (E12.5 liver-1), ENCFF144DHB (E12.5 liver-2), ENCFF971KKK (E13.5 liver-1), ENCFF042DVY (E13.5 liver-2), ENCFF770SOB (E10.5 heart-1), ENCFF351QKG (E10.5 heart-2), ENCFF159DWP (E11.5 heart-1), ENCFF168UJM (E11.5 heart-2), ENCFF484QWQ (E12.5 heart-1), ENCFF329HOZ (E12.5 heart-2), ENCFF148BEQ (E13.5 heart-1), ENCFF836QQS (E13.5 heart-2), ENCFF145PTV(E10.5 forebrain-1), ENCFF476ADM (E10.5 forebrain-2), ENCFF606UHO (E11.5 forebrain-1), ENCFF434CSI (E11.5 forebrain-2), ENCFF928MQD (E12.5 forebrain-1), ENCFF046RSQ (E12.5 forebrain-2), ENCFF960KJV (E13.5 forebrain-1), ENCFF356CTG (E13.5 forebrain-2). Placenta RNA transcript abundance was obtained from Soncin et. al. (GSE100053). Expression data were filtered based on known metabolic genes47-49 and human-mouse gene mapping was based on the HomoloGene database (https://www.ncbi.nlm.nih.gov/homologene).

# Field-specific reporting

Please select the one below that is the best fit for your research. If you are not sure, read the appropriate sections before making your selection.

☒ Life sciences      ☐ Behavioural & social sciences      ☐ Ecological, evolutionary & environmental sciences

For a reference copy of the document with all sections, see nature.com/documents/nr-reporting-summary-flat.pdf

# Life sciences study design

All studies must disclose on these points even when the disclosure is negative.

| Sample size | Samples sizes were not pre-determined based on statistical power calculations but were based on our experience with these assays. For most experiments, the minimum number of mice was 3, with some exceptions where the embryo/placenta numbers were n≥10. For metabolomics and isotope tracing studies in mice, our previous publications (PMID: 31853067 PMCID: PMC6930341, PMID: 28985563 PMCID: PMC5684706) have determined that 3-5 mice is a sufficient sample size to obtain statistical power. Because our statistics was performed on placenta and embryo numbers that exceed the number of mice in the study, we determined that using 3 mice would be sufficient as biological replicates unless the number of placenta/embryos obtained from 2 mice was equivalent to experiments where 3 mice were used. This number was typically n>10. |
|---|---|
| Data exclusions | No data were excluded; however, sometimes the small sample size was below the threshold for metabolomic analysis. In those instances, data that could be obtained from maternal blood or other tissues was used. These samples were not used during direct comparisons of embryo relative to its own placenta if one of the samples was absent. As well, any conceptus that was observed to contain hemorrhage or necrotic tissue was deemed "non-viable" and not used for any additional studies. |
| Replication | The experimental findings were reproduced in multiple independent experiments. The number of independent experiments and biological replicates for each data panel is indicated in the figure panel itself or in the figure legends, and in the source data files. Data shown in the figures represent the aggregate of all independent experiments in most cases.  Data shown in a minority of panels are from a representative experiment (e.g. for histology) and in those cases the number of independent experiments that reproduced the finding is also indicated in the figure legends. For patient data, we show cell counts that were included in the single patients chart over her lifetime. |
| Randomization | No formal randomization techniques were used; however, samples were allocated randomly to experiments and processed in an arbitrary order. |
| Blinding | During flow cytometry, isotope tracing, metabolomics, qPCR, tissue weights, somite counts  and histology experiments, the data were analyzed in a manner blinded to sample genotype. A.S. collected the samples and then passed them to A.T. for flow cytometry, or to I.M-M. and M.A.C. for histology and immunofluorescence, and A.T. for qPCR. A.S. processed samples for mass spectrometry and analyzed data. After the patterns had been analyzed in each of these experiments, D.D. provided the genotype information so results could be interpreted. For experiments in wild-type mice, no blinding was performed on placentas versus embryos because A.S. performed these experiments and analyzed the data. For gene expression studies from publicly available datasets, no blinding was performed. |

# Reporting for specific materials, systems and methods

We require information from authors about some types of materials, experimental systems and methods used in many studies. Here, indicate whether each material, system or method listed is relevant to your study. If you are not sure if a list item applies to your research, read the appropriate section before selecting a response.

## Materials & experimental systems

| n/a | Involved in the study |
|-----|----------------------|
| ☐ | ☒ Antibodies |
| ☒ | ☐ Eukaryotic cell lines |
| ☒ | ☐ Palaeontology and archaeology |
| ☐ | ☒ Animals and other organisms |
| ☐ | ☒ Human research participants |
| ☐ | ☒ Clinical data |
| ☒ | ☐ Dual use research of concern |

## Methods

| n/a | Involved in the study |
|-----|----------------------|
| ☒ | ☐ ChIP-seq |
| ☐ | ☒ Flow cytometry |
| ☒ | ☐ MRI-based neuroimaging |

# Antibodies

| | |
|---|---|
| Antibodies used | The following antibodies have been used in this study:<br><br>Anti-Mouse Ter119 APC<br>clone: TER-119<br>REF 20-5921-U100<br>LOT C5921040320203,<br>TONBO<br>1:100 Flow<br><br>Anti-Mouse CD71 FITC<br>clone R17217 (R17 217.1.4)<br>LOT 2213055<br>REF 2213055<br>Biolegend<br>1:100 Flow<br><br>Anti-Mouse CD41 PE/Cy7<br>clone: MWReg30<br>cat 133916<br>LOT B27086<br>Biolegend<br>1:100 Flow<br><br>Anti-Mouse CD117 (c-KIt) APC-eFluor 780<br>clone: 2B8<br>Ref: 47-1171-82<br>LOT 2261910<br>Invitrogen<br>1:100 Flow<br><br>Rat-anti-PECAM<br>clone: MEC 13.3 (RUO)<br>BD, Biosciences,<br>cat #553370<br>1:100 Whole mount IF<br><br>Rat-anti-Endomucin<br>clone: V.7C7<br>Santa Cruz, sc-65495<br>1:100 Whole mount IF<br><br>Rabbit-anti-Connexin 40<br>Alpha Diagnostics International, cat#CX-40A<br>1:100 Whole mount IF<br><br>donkey-anti-rat 488<br>Invitrogen, cat #A21208<br>1:250 secondary Whole mount IF<br><br>donkey-anti-rabbit 555<br>Invitrogen, cat# A31572<br>1:250 secondary Whole mount IF |
| Validation | All antibodies are commercially available and have been validated in previously published studies<br><br>Anti-Mouse Ter119 APC:<br>Egusquiza RJ, Ambrosio ME, Wang SG, Kay KM, Zhang C, Lehmler HJ, Blumberg B. Evaluating the Role of the Steroid and Xenobiotic |

Receptor (SXR/PXR) in PCB-153 Metabolism and Protection against Associated Adverse Effects during Perinatal and Chronic Exposure in Mice. Environ Health Perspect. 2020 Apr;128(4):47011. doi: 10.1289/EHP6262. Epub 2020 Apr 30.

Anti-Mouse CD71 FITC:
Tsai S, Clemente-Casares X, Zhou AC, Lei H, Ahn JJ, Chan YT, Choi O, Luck H, Woo M, Dunn SE, Engleman EG, Watts TH, Winer S, Winer DA. Insulin Receptor-Mediated Stimulation Boosts T Cell Immunity during Inflammation and Infection. Cell Metab. 2018 Dec 4;28(6):922-934.e4. doi: 10.1016/j.cmet.2018.08.003. Epub 2018 Aug 30. PMID: 30174303.

Anti-Mouse CD41 PE/Cy7:
Gentek R, Ghigo C, Hoeffel G, Bulle MJ, Msallam R, Gautier G, Launay P, Chen J, Ginhoux F, Bajénoff M. Hemogenic Endothelial Fate Mapping Reveals Dual Developmental Origin of Mast Cells. Immunity. 2018 Jun 19;48(6):1160-1171.e5. doi: 10.1016/j.immuni.2018.04.025. Epub 2018 May 29. PMID: 29858009.

Anti-Mouse CD117 (c-KIt) APC-eFluor 780:
Di Genua C, Valletta S, Buono M, Stoilova B, Sweeney C, Rodriguez-Meira A, Grover A, Drissen R, Meng Y, Beveridge R, Aboukhalil Z, Karamitros D, Belderbos ME, Bystrykh L, Thongjuea S, Vyas P, Nerlov C. C/EBPα and GATA-2 Mutations Induce Bilineage Acute Erythroid Leukemia through Transformation of a Neomorphic Neutrophil-Erythroid Progenitor. Cancer Cell. 2020 May 11;37(5):690-704.e8. doi: 10.1016/j.ccell.2020.03.022. Epub 2020 Apr 23. PMID: 32330454; PMCID: PMC7218711.

Rat-anti-PECAM:
Baldwin HS, Shen HM, Yan HC, et al. Platelet endothelial cell adhesion molecule-1 (PECAM-1/CD31): alternatively spliced, functionally distinct isoforms expressed during mammalian cardiovascular development. Development. 1994; 120(9):2539-2953. (Clone-specific: Blocking).

Rat-anti-Endomucin:
Wang, C.|Ying, J.|Nie, X.|Zhou, T.|Xiao, D.|Swarnkar, G.|Abu-Amer, Y.|Guan, J.|Shen, J.| et al. 2021. Bone Res. 9: 29.PMID: # 34099632

Rabbit-anti-Connexin 40
Shekhar A, Lin X, Liu F, Zhang J, Mo H, Bastarache L, et al. Transcription factor ETV1 is essential for rapid conduction in the heart. J Clin Invest. 2016;126:4444-4459

# Animals and other organisms

Policy information about studies involving animals; ARRIVE guidelines recommended for reporting animal research

| Laboratory animals | All mice were housed in a pathogen free environment (Temperature 68-79F, Humidity: 30-70%) with a 12:12 light/dark cycle and fed chow diet ad libitum.  Wild-type C57/BL6 females and males were obtained either from Jackson Labs or from the UT-Soutwestern Breeding Core Facility. Males were only used to initiate pregnancies and were between 3-6 months old. Females were 8-12 weeks old and naively pregnant.<br><br>LIPT1 N44S mice were generated at the Children's Research Institute (CRI) Mouse Genome Engineering Core as previously described (Ni M, et. al.  Functional Assessment of Lipoyltransferase-1 Deficiency in Cells, Mice, and Humans. Cell Rep. 2019 Apr 30;27(5):1376-1386.e6. doi: 10.1016/j.celrep.2019.04.005. PMID: 31042466; PMCID: PMC7351313.) These mice were back crossed to C57/BL6 mice for at least 10 generations and  maintained by breeding either as heterozygous breeding pairs or crossed to C57/BL6. Gender specific breeding was not used. Female LIPT1 WT/N44S pregnant mice (8-15 week) were used for experiments. |
|---|---|
| Wild animals | no wild animals were used in these studies. |
| Field-collected samples | no field-collected samples were used in these studies |
| Ethics oversight | All procedures were approved by the UT Southwestern Animal Care and Use Committee (IACUC) in accordance with the Guide for the Care and Use of Laboratory Animals. |

Note that full information on the approval of the study protocol must also be provided in the manuscript.

# Human research participants

Policy information about studies involving human research participants

| Population characteristics | This manuscript describes a single female patient diagnosed with LIPT1 deficiency (genotype: LIPT1: c.875C > G (p.S292X),c.131A > G (p.N44S))  that has been followed since shortly after birth to a current age of 16. |
|---|---|
| Recruitment | This patient was recruited as part of a larger study (NCT02650622). |
| Ethics oversight | All subjects were enrolled in the study (NCT02650622) approved by the Institutional Review Board (IRB) at University of Texas Southwestern Medical Center (UTSW). Informed consent was obtained from all patients and their families. |

Note that full information on the approval of the study protocol must also be provided in the manuscript.

# Clinical data

Policy information about clinical studies

All manuscripts should comply with the ICMJE guidelines for publication of clinical research and a completed CONSORT checklist must be included with all submissions.

| | |
|---|---|
| Clinical trial registration | NCT02650622 |
| Study protocol | https://clinicaltrials.gov/ct2/show/NCT02650622 |
| Data collection | n/a |
| Outcomes | n/a |

# Flow Cytometry

## Plots

Confirm that:

☒ The axis labels state the marker and fluorochrome used (e.g. CD4-FITC).

☒ The axis scales are clearly visible. Include numbers along axes only for bottom left plot of group (a 'group' is an analysis of identical markers).

☒ All plots are contour plots with outliers or pseudocolor plots.

☒ A numerical value for number of cells or percentage (with statistics) is provided.

## Methodology

| | |
|---|---|
| Sample preparation | Whole embryos were dissected and dissociated in cold-PBS using a disposable pestle (VWR). To obtain a single-cell suspension, cells were filtered through a 40 um cell strainer and then stained with the appropriate antibodies. |
| Instrument | BD Fortessa (for analysis) |
| Software | BD FACSDiva 8.0, FlowJo V10 |
| Cell population abundance | 30,000 cells were sorted from each embryo. CD71-/TER119- cells were 10% of the total population, CD71+/TER119+ (Erythrocytes) cells were 0.5-1% of the total population. CD71-/TER119-/CD41+ (MEPs) cells were 0.2-0.7% of the population. |
| Gating strategy | To eliminate dead cells from analyses, cells were stained with 4',6-diamidino-2-phenylindole (DAPI). Mouse erythrocytes were identified as being positive for endothelial markers Ter119 and CD71. Megakaryocyte and erythroid progenitor cells were identified as negative for TER119, CD71, and CD117 (c-KIT) but positive for CD41. |

☒ Tick this box to confirm that a figure exemplifying the gating strategy is provided in the Supplementary Information.

