## [Peer Review File · Nature]

Manuscript Title: Compartmentalized metabolism supports midgestation mammalian development

Reviewer Comments & Author Rebuttals

Reviewer Reports on the Initial Version:

Referee #1 (Remarks to the Author):

Solmonson et al. studies the metabolism of mouse embryo and placenta during midgestation (day 9.5 to 13.5). Notably, the study was done in vivo and by means of methods that allow distinguishing the two "compartments". It builds on an impressive number of tedious and technically difficult tracing experiments, which provided an unprecedented view on the gestational process. The work is divided on four parts: (I) a metabolomics analysis that highlights a separation between embryo and placenta, and its timing; (II) a ¹³C-tracing analysis on intracellular glucose utilization, with a particular focus on pentose phosphate shunt and purine/pyrimidine biosynthesis; (III) ¹³C-tracing and expression analyses pivoting on mitochondrial oxidation; (IV) an analysis of animals with impaired lipoylation and, thus, mitochondrial oxidative activity. This caused embryonic lethality after gd 10.5 in spite of normal glucose uptake and breakdown to glycolysis. Overall, this study indicates fundamental and essential metabolic differences and trends between placenta and embryo during midgestation.

Major points:

- The discussion of the metabolome data is biased (figure 1). Firstly, the most striking changes are between placenta and embryo samples - regardless of the gestational day (figure 1cd) - but the authors focus on the transition between gds (figure 1efg). Given that the whole manuscript is about metabolic differences between embryo and placenta, the striking "baseline" differences should be properly discussed before trends. Secondly, the majority of metabolite differences are seen in the placenta, with more significant enrichments, but the attention is directed to the embryo. Thirdly, it is not clear why the authors single out purines and pyrimidines when there are many more pathways (i.e. amino acids) affected and enriched. I understand that the authors are trying to channel the discussion into aspects they like, but there is a strong bias.

- A major claim of the study is that glucose fate is different between embryo and placenta. As it stands now, this is not supported by the data. There are obvious differences in ¹³C enrichment (figure 2hijk), but they all could be a mere consequence of the delayed or lower G6P labeling (fig 2e). There are alternative explanations for the differences in G6P that do not imply differences in fluxes in primary metabolism. For instance, it could be that the difference is upstream of G6P (e.g. increased glycogenolysis vs. glucokinase in the placenta), or that the intracellular G6P concentration is larger in placenta causing slower enrichment kinetics regardless of fluxes. More is needed to safely conclude that glucose catabolism or purine synthesis differs between the two compartments.

- The time-resolved tracing experiment (figure 3) is used to claim that glucose metabolism in embryo drifts, while in placenta it remains constant or follows an inverted trend. This is solid, but it should be noted that the ¹³C labeling of the two compartments converge to similar values (figure 3c-e). Intuitively, this would indicate that their metabolic activity becomes similar during gestation, in sharp contrast with the claimed divergence of the embryo. Please clarify.

- The use and relevance of the hypomorphic LIPT1 mutation require more explanations. First, disrupting LIPT1 has catastrophic consequences that reach well beyond glucose oxidation. The question is whether this experiment is specific enough to claim that glucose oxidation is essential.

I would argue that inhibiting ETC or preventing pyruvate transport in mitochondria would have been a much more surgical intervention to prove the relevance of embryonic glucose oxidation during midgestation. Second, there is no data to prove that placental metabolism is not – or less - affected by the LIPT1 mutation. Supp figure 4d suggests the opposite. This seems an important control to claim compartmentalization of glucose oxidation, or that placenta function is normal (line 37 of the abstract).

Minor points:

- Figure 1d: I am puzzled by the normalization of metabolomics data. If the data is autoscaled, there is no reason of using log2 on top of z-scores. In fact, it skews the data in favor of subtle changes.
- Lines 93-95: the lower enrichment of G6P or F6P cannot be justified by exchange with trioses in lower glycolysis. Beyond generating unlabeled G6P/F6P, it would also produce m+4 molecules with equal likelihood. Since m+4 is not detectable (Supp Figure 2b), the hypothesis is wrong.
- Lines 128 and following: the ratio "Pyruvate enrichment / citrate enrichment" does not inform on pyruvate oxidation (i.e. on how much of the pyruvate is oxidized), but on how much citrate originated from pyruvate. It should not be used as a proxy for the amount of pyruvate that is oxidized in mitochondria.
- Line 192: "The data indicate that lipoylation facilitates a precisely-timed change in mitochondrial metabolism required for development past gd10.5." reads like a change in lipoylation drives the metabolic changes in glucose and is responsible for its timing. There is no evidence for this. Timing could be simply triggered by oxygenation. Please correct.

Referee #2 (Remarks to the Author):

The manuscript by Solmonson et al. entitled "Compartmentalized glucose oxidation supports midgestation mammalian development" describes a novel approach to directly assess "evolving" metabolic transitions in utero in model organisms like mice. Isotope tracing and metabolomics identify evolving metabolic programs in both the placenta and embryo over gestational days (gd)10.5 to 13.5. Interestingly, gd10.5 to gd11.5 mark a period of significant divergence between placenta and embryo proper. For example, glucose carbons' contribution to the TCA cycle increases through midgestation in the embryo but not the placenta. Subsequently, the embryo contains organ specific (e.g. heart, brain, liver) metabolic programs that include pyruvate oxidation in some cases. Finally, the authors describe a lipoyltransferase-1 (LIPT1) mutant strain of mice and demonstrate that this results in decreased glucose oxidation in both embryo and placenta leading to lethality at gd11.5. These findings serve as a functional confirmation of the importance enhanced glucose oxidation and other fuels at midgestation. Overall, the methods described are highly innovative and dependent on significant surgical skills, precise metabolomics, and metabolic labeling approaches. I recommend acceptance of the paper upon resolution of the following concerns:

1. Data displayed in Figure 1d indicates that placenta at gd11.5 have a striking decrease in a large number of metabolites that resolves just one day later. Can the authors account for this observation?
2. From gd9.5 to gd12.5, the embryos exhibit more dramatic increases in mass than the placenta. Do the authors believe that the distinct metabolic programs of these two tissues support this phenotype?
3. Later in the manuscript, the authors rely on transcriptomics to examine glucose oxidation in fetal heart and brain, which revealed increased ETC-related transcripts particularly in the heart. By the same argument, could transcriptomics in the placenta versus the embryo explain the dilution by placental labeled glucose-6-phosphate due to glycogenolysis, gluconeogenesis, and other pathways? While RNA abundance can't explain everything, of course, it could also be another way

to examine this issue as described in Figure 3f.

4. The metabolomics and metabolic labeling indicate a dramatically increased expansion in the purine pool (see Figure 1f and g). The labeling scheme depicted can also detect pyrimidines? If so, why was this not performed. One wonders if this activity is to support DNA synthesis or as bioenergetics (via ADP) or both.

Referee #3 (Remarks to the Author):

This is an interesting study with the ambitious goal of elucidating the metabolism of placenta and embryo during a key phase of organogenesis and embryonic development using mice as model system. The authors found that around gd10.5-11.5 the metabolism, specifically glucose metabolism, starts to compartmentalise between embryo and placenta. The required methodologies were challenging and involved in vivo assessment of metabolic activity, infusion of isotope-labeled metabolites, and testing of a mouse strain deficient in glucose oxidation.

The labeled metabolite tracing experiments with wild-type mice are mostly clear and convincing (with a few comments pointed out below). But I'm not convinced that the *Lipt1N44S* mice are a suitable model to study metabolism at gd10.5 when homozygotes die at the gd10.5-gd11.5 transition. The authors need to clarify and show that results are not confounded by the imminent death of embryos (specific concerns are pointed out further below).

Detailed comments:

Figure 1

Line 68: "Unsupervised clustering of metabolomics data indicated that placenta and embryo differ from each other on all days examined (Figure 1c,d).

Please clarify this statement because the PCA plot (Figure 1c) disagrees with this statement.

Metabolites in the E13.5 placenta samples cluster inside E12.5 placenta samples. There is a large spread between samples at E12.5, which would suggest that E13.5 placentas are similar.

Figure 2

Supplemental figure 2a is helpful to understand the glucose labelling results. I suggest putting it into the main text and swapping it with the current Figure 2a which can be moved to the supplement (the procedure is described in the text).

Infusion and serial dissection of conceptuses over 3 hours (Figure 2 and page 4, line 80 onwards): The idea of doing a kinetic analysis by serially dissecting conceptuses over a time while infusing labelled metabolites is good, however, appropriate controls are lacking. It is possible that the longer the mother is under anesthesia with repeated exposure to the procedure that maternal metabolism might be affected. Controls would be (i) maintaining the anesthesia and infusion over 3 hours but only dissecting once at the end of the 3 hours; (ii) the effect of the anaesthetic can be assessed by dissecting control litters at the same embryonic stage in which the mother is culled before dissection.

Line 116: "Like R5P, serine and glycine were extensively labeled in the embryos, indicating the rapid production of multiple purine precursors (Figure 2k)."

Check this statement as Fig2k shows no difference in R5P between embryo and placenta.

LIPT1

The embryo/placenta data should be represented and discussed as *Lipt1+/+*, *Lipt1+/N44S* or *Lipt1N44S/N44S* and not as healthy and mutant. If *Lipt1+/+*, *Lipt1+/N44S* are grouped in the graphs, then colour code the data points so it is clear which represent the two different genotypes.

Fig. 4a: About half of the mutant embryos appear to be about 8 fold lighter (smaller) than the average weight of healthy embryos, whereas the placentas have similar weights between phenotypes. I am concerned that the metabolic data from these severely affected tiny embryos reflect general embryo demise rather than the specific effect of the mutation on metabolites especially given that homozygotes “undergo embryonic lethality at the gd10.5-gd11.5 transition” (line 164). Was any embryo lethality observed at gd10.5 in the litters used to generate these data? If so, the number of dead embryos should be stated per litter.

Line 164: The authors say: “Lipt1N44S/+ mice are healthy and born at the expected ratio and their metabolomic signatures are similar to Lipt1+/+ tissues at gd10.5. (Supplemental Figure 4a,b)”

The authors should improve the clarity of this paragraph and explain Supplemental Figure 4a and b better. Especially because this is a key experiment from which they drew the conclusion that both wild-types and heterozygotes are “healthy” and can be pooled for analyses. Specifically, which metabolic signatures were similar and different? Is this conclusion based on the embryo plot alone, because the separation appears less clear-cut in the placenta?

Line 167: Lipt1N44S/N44S conceptuses are small but viable at 10.5.

This should be discussed further: Are they small because they are developmentally delayed? Then the metabolic profiles at gd10.5 would be different to heterozygotes and wild-types because they resembled those of earlier conceptuses. Or are they small due to a lack of cell division, but metabolically matured like a normal gd10.5 conceptus? Since the authors did not see difference in expression markers in placentas between genotypes, and placenta weights were similar, it suggests that the Lipt1N44S/N44S placenta has developed normally.

The authors could answer this by doing a differentiation marker expression analysis (Fig S4) with embryos, as done for the placentas. It would show if the small gd10.5 embryos have similar gene expression and therefore are indeed developed to gd10.5.

On a related note, the authors should discuss what they consider the cause and effect of the observed differences in Lipt1N44S/N44S embryos. Are they small at gd10.5 because of the metabolic perturbation, i.e cell division and maturation are impaired because of the limited energy generation via TCA cycle, etc? Or do the embryos have metabolic changes as a result of abnormal embryonic development, e.g. metabolically active cell types are underrepresented due to perturbed embryonic development?

Figure 4C, D, E, F; and Supplemental Figure 4C:

Metabolite levels and expression of placental layer markers are shown as fold change from healthy placentas/embryos. How exactly was this calculated? These graphs have datasets for healthy embryos/placentas whose data points are scattered and have averages that differ from 1 (e.g. Amino adipate in Fig. 4F). A different dataset of healthy tissues must have been used for normalising and determining the fold changes. Please explain.

Conclusion

“Genetically-defined metabolic defects can result in congenital anomalies in humans, emphasizing the importance of precise metabolic control during fetal development.” After this sentence, I expected some discussion about the relevance of this manuscript’s findings for human health and disease, but none is given. Human case reports of LIPT1 mutation exist and it might be worthwhile mentioning them? In addition, biallelic variants of the upstream genes LIPT2 and LIAS are described and lead to disease phenotypes.

I would have liked to see some discussion about how the findings of this paper translate to humans, their relevance for clinical intervention, etc.

“We report highly compartmentalized metabolic phenotypes in the placenta and embryo, with both tissues undergoing extensive but largely private metabolic changes between gd10.5-gd11.5. The placenta transfers glucose and other nutrients from the maternal circulation to the embryo, but

also displays localized glucose turnover. Surprisingly, these aspects of placental glucose metabolism have little impact on glucose metabolism in the embryo, which is supplied almost exclusively by glucose delivered from the maternal blood.”

Very interesting study but one should not be surprised that the embryo and placenta have distinct metabolic profiles as they are comprised of very distinct types and numbers of cells, with the placenta having relatively few cell types. Perhaps this point could be added to the discussion.

Methods

Line 239: Why were the homogenised samples for metabolomic analysis subjected to three freeze-thaw cycles with liquid nitrogen before BCA analysis and LC-MS or GC-MS? Freeze-thaw cycles are considered to be detrimental to proteins in solution and might also affect the analytes of interest. Have the authors checked the impact of the 3-times freeze-thawing on BCA assay values and the measured metabolites?

Minor comments:

ETC = electron transport chain. The abbreviation should be explained at the first appearance in the text

Figure 1e: Abbreviation FC should be explained

Supplemental figure 3: Please provide a brief explanation what the abbreviations Ndufa, SDHb, Cyc1, etc. stand for. Readers outside the field may not be immediately familiar with these genes.

Line 240: “centrifuged at top speed”. Please provide proper centrifugal force value or rpm and model of centrifuge.

Author Rebuttals to Initial Comments:

We thank the editors and Reviewers for the comments and interest in our manuscript. A point-by-point response to the referee comments follows below. *Our responses are shown in italics.*

Referees' comments:

Referee #1 (Remarks to the Author):

Solmonson et al. studies the metabolism of mouse embryo and placenta during midgestation (day 9.5 to 13.5). Notably, the study was done in vivo and by means of methods that allow distinguishing the two “compartments”. It builds on an impressive number of tedious and technically difficult tracing experiments, which provided an unprecedented view on the gestational process. The work is divided on four parts: (I) a metabolomics analysis that highlights a separation between embryo and placenta, and its timing; (II) a ¹³C-tracing analysis on intracellular glucose utilization, with a particular focus on pentose phosphate shunt and purine/pyrimidine biosynthesis; (III) ¹³C-tracing and expression analyses pivoting on mitochondrial oxidation; (IV) an analysis of animals with impaired lipoylation and, thus, mitochondrial oxidative activity. This caused embryonic lethality after gd 10.5 in spite of normal glucose uptake and breakdown to glycolysis. Overall, this study indicates fundamental and essential metabolic differences and trends between placenta and embryo during midgestation.

Major points:

1. The discussion of the metabolome data is biased (figure 1). Firstly, the most striking changes are between placenta and embryo samples - regardless of the gestational day (figure 1cd) - but the authors focus on the transition between gds (figure 1efg). Given that the whole manuscript is about metabolic differences between embryo and placenta, the striking “baseline” differences should be properly discussed before trends. Secondly, the majority of metabolite differences are seen in the placenta, with more significant enrichments, but the attention is directed to the embryo. Thirdly, it is not clear why the authors single out purines and pyrimidines when there are many more pathways (i.e. amino acids) affected and enriched. I understand that the authors are trying to channel the discussion into aspects they like, but there is a strong bias.

RESPONSE: We agree that there are many differences between the embryo and placenta on every day examined. This is not unexpected because of the divergent cellular composition and functions between these tissues, but we provide more text to emphasize the metabolic differences and point out pathways that most differ between placenta and embryo on gd10.5 (see new text below). In addition to providing the full metabolomics dataset for readers specifically interested in metabolic differences between placenta and embryo, we revised supplemental figure 1. This figure now includes a heatmap of the 50 metabolites that differ most significantly between embryos and placentas (Supplemental Figure 1b), a list of metabolites that contribute to profile separation (Supplemental Figure 1c) and an analysis of pathways that are significantly changed in the placenta between gd10 and gd11 (Supplemental Figure 1d), in contrast with the significant pathways observed in the embryo at the same time (Figure 1f). We hope this eliminates concerns about bias in the presentation, because this figure now explicitly highlights differences between embryo and placenta.

We also agree that the metabolic analysis revealed many pathways that would be worth further study. But as is always the case when presenting a large amount of data acquired using a new method, we needed to make choices about where to focus attention for the subsequent experiments. Part of the rationale for developing this technique was to understand evolving metabolic activities during midgestation and their relevance to developmental defects that appear in patients with inborn errors of metabolism. For that reason, much of the subsequent attention in the paper is devoted to the embryo. Along these lines, the changes in purines and pyrimidines are interesting because a) these pathways are involved in cell growth and proliferation but understudied in the context of embryonic development; b) perturbation of these pathways in utero can cause developmental defects in humans¹⁻³; c) purine and pyrimidine synthesis integrate many other pathways, including the pentose phosphate pathway and nonessential amino acid synthesis; and d) the metabolomics data document marked alterations in these pools in both tissues, particularly an abrupt and sustained expansion of purine metabolites in the embryo after gd10.5 (Figure 1g). For these reasons, we felt that focusing some attention to nucleotide metabolism would be interesting and justified by the data. The amino acid-related pathways are also interesting, and we now mention them in the text. But please note that KEGG lists distinct pathways for many individual amino acids, producing many “hits” on MSOA. In contrast, the

purines and pyrimidines are consolidated into one pathway each, thereby underrepresenting the changes in these metabolites by MSOA.

New text starting at line 77:

“Placenta and embryo metabolomics differ throughout midgestation, as expected given their divergent cellular composition and functions (Figure 1c,d, Supplemental figure 1a). In both tissues, gd10.5 was metabolically distinct from subsequent days, indicating transitions between gd10.5 and gd11.5 (Figure 1d, Supplemental Figure 1a). These transitions were largely private, with most metabolites changing in one tissue but not the other (Figure 1e,f Supplemental Figure 1b-d). Metabolic set overrepresentation analysis (MSOA) identified numerous pathways that change in the placenta between gd10.5-gd11.5, particularly pathways related to nitrogen and amino acid metabolism (Supplemental Figure 1d). Urea cycle-related metabolites increased abruptly but transiently in the placenta at gd11.5 (Supplemental Figure 1e), possibly reflecting arginine’s role in stimulating placental-fetal blood flow⁴. In the embryo, MSOA between gd10.5-gd11.5 identified purine and pyrimidine metabolism as two of the top-scoring pathways (Figure 1f). Most purines displayed a sustained increase after gd10.5 in the embryo, whereas pyrimidines showed little change or decreased in both tissues (Figure 1g).”

2. A major claim of the study is that glucose fate is different between embryo and placenta. As it stands now, this is not supported by the data. There are obvious differences in ¹³C enrichment (figure 2hijk), but they all could be a mere consequence of the delayed or lower G6P labeling (fig 2e). There are alternative explanations for the differences in G6P that do not imply differences in fluxes in primary metabolism. For instance, it could be that the difference is upstream of G6P (e.g. increased glycogenolysis vs. glucokinase in the placenta), or that the intracellular G6P concentration is larger in placenta causing slower enrichment kinetics regardless of fluxes. More is needed to safely conclude that glucose catabolism or purine synthesis differs between the two compartments.

RESPONSE: The Reviewer raises concerns about interpretation of the labeling data and we are grateful for the opportunity to clarify. We should have been more precise in this paragraph. The Reviewer is right that a difference in G6P labeling from ¹³C-glucose does not necessarily mean that the rate of glucose metabolism was different; it could also mean that alternative sources of G6P (unlabeled, or labeled differently than G6P arising directly from ¹³C-glucose) differed between the tissues. Therefore the findings related to G6P are better referred to as differences in carbohydrate metabolism rather than differences in glucose metabolism, and this is now clarified in the text (line 107). More detailed experiments, ideally involving genetic models that would allow us to selectively inactivate pathways in the placenta or embryo, are needed to fully explain these labeling differences. For this reason, we are circumspect about why G6P labeling differs between the tissues, although we do mention some possibilities. We also do not make any claims about quantitative flux differences. Nevertheless, the important conclusion from these experiments is that there are metabolic differences between placenta and embryo reflected in changes in metabolite labeling. Altered G6P labeling is one example of this, but it is not the only one. Several other aspects of metabolism unrelated to G6P labeling, including the relative labeling between TCA cycle intermediates and pyruvate, also differ between placenta and embryo. Again, the point we want to communicate to the reader is that the labeling approach is sensitive enough to detect such differences.

We took several measures, including editing the text, adding new data and simplifying the presentation, to highlight metabolic differences between placenta and embryo and to clarify what is still unknown:

- We added a new panel (Figure 2d) to illustrate some of the pathways that may contribute to labeling in G6P and related metabolites.*
- We removed the experiments with [1,2-¹³C]glucose and focused on labeling with [U-¹³C]glucose. We think this will make it easier to follow the key points.*
- As stated above, the fact that G6P labeling differs so much between embryo and placenta is evidence of altered carbohydrate metabolism between the tissues, even if the explanation is upstream of G6P (this is now illustrated in Figure 2d). We agree that this could be related to glycogenolysis in the placenta, as has been argued elsewhere in the literature^{5,6}. However, we do not think that the labeling differences can be explained simply by a large placental G6P pool that acquires label less rapidly than in the embryo, because the kinetics show the two compartments acquiring label at approximately the same time, with steady-state labeling appearing earlier if anything in the placenta (Figure 2e). This pattern is more consistent with the involvement of unlabeled or alternatively labeled G6P, especially in*

the placenta. We think this is an interesting observation, although we acknowledge that definitive delineation of all sources of placental G6P will require additional genetic models. We edited the text as follows:

New text starting at line 100:

“Rapid labeling of downstream metabolites indicates robust metabolism in the conceptus, and distinct labeling features in the embryo and placenta indicate metabolic differences between the tissues. ^{13}C enrichment in glucose-derived metabolites reflects the combined contribution of labeled and unlabeled substrates through intersecting pathways (Figure 2d). Glucose-6-phosphate (G6P) appeared rapidly in the embryo as m+6, indicating conversion from maternal glucose (Figure 2e). However placental G6P was labeled differently in the same mice. Overall G6P enrichment was lower than in the embryo, and G6P m+6 and m+3 appeared over similar time scales (Figure 2e). A complete understanding of carbohydrate metabolism will require compartment-specific enzyme knockouts, but the placental labeling pattern suggests contributions from glycogenolysis, gluconeogenesis and other pathways previously reported in mammalian placentas (Figure 2d)^{5,6}.”

- *Similarly to G6P, R5P labeling also indicates differences between placenta and embryo. Total enrichment (i.e. the unlabeled fraction subtracted from the total) is similar between the tissues, but the isotopologue distribution is different (Figure 2f, Supplemental Figure 2b). In particular, fully labeled R5P (m+5) is lower in the placenta. This suggests that each tissue synthesizes R5P from labeled glucose, but the pathways contributing to R5P differ.*
- *In purines, total labeling is higher in the embryo, and the isotopologue distribution is very similar to what is observed in R5P (m+1 through m+5, see Figure 2f and Supplemental Figure 2c-e). Therefore most labeling in purines is derived from the ribose group. Importantly, however, in the three purine nucleotides where signal-to-noise was adequate to assess isotopologue distributions, a fraction of the pools contained higher-order labeling (e.g. m+6 and higher) indicating that part of the purine ring was also labeled (Supplemental Figure 2c-e). These labeled species tended to be higher in the embryo and barely detectable in the placenta. These data indicate that a) in both the embryo and placenta, R5P and purine nucleotides turn over rapidly so that a high fraction contains ^{13}C after just a few hours; b) it is likely that both tissues use a combination of de novo nucleotide synthesis and salvage; and c) evidence of de novo synthesis (i.e. labeling higher than m+5) is stronger in the embryo, where purine pools are also expanding over this period. We believe these labeling patterns support the argument that metabolic differences exist between placenta and embryo on gd10.5. We emphasize, though, that rates of turnover are very high in both tissues.*
- *As an orthogonal approach to assess purine metabolism, we performed new isotope tracing experiments using [γ - ^{15}N]glutamine (i.e. glutamine labeled with ^{15}N in the amide position), which is incorporated directly onto the purine ring during de novo synthesis. Here we again observed substantial purine labeling in both tissues, with higher IMP and GMP labeling in the embryos (Figure 2g). Because [γ - ^{15}N]glutamine reports nitrogen incorporation into nucleotides independently of the carbon contributions arising from G6P, this labeling experiment provides independent evidence of enhanced purine synthesis in embryos. We edited the text discussing R5P and nucleotide labeling as follows:*

New text starting at line 111:

“The pentose phosphate pathway intermediate and nucleotide precursor ribose-5-phosphate (R5P) also turned over rapidly. R5P is similar to G6P in that labeling was distributed across several isotopologues, and fully-labeled R5P (m+5) was the predominant labeled form in the embryo but not the placenta (Supplemental Figure 2a,b). After 4 hours, purines were extensively labeled in both placenta and embryo, but again labeling was higher in embryos (Figure 2f, Supplemental Figure 2c-e). The total enrichment (i.e. [1.0 – unlabeled fraction], incorporating ^{13}C entry from all sources) was over 0.3 in all purines analyzed, indicating that within 4 hours, at least 30% of the embryo purine pools contained carbon originating in the maternal circulation (Figure 2f). Although much of the purine labeling appeared to arise from R5P, purine bases in the embryo also contained ^{13}C ; evidence for labeling in the bases included higher total labeling in purines than R5P (Figure 2f), and the presence of IMP, GMP and AMP containing more than five ^{13}C nuclei (Supplemental Figure 2c-e). Taken in the context of the expanding purine pool (Figure 1g), and extensive labeling of serine and glycine (Supplemental Figure 2f), these data point to de novo purine synthesis in embryos. As an orthogonal labeling approach, we infused

pregnant mice with [γ - ^{15}N]glutamine, which is incorporated into the purine ring during de novo synthesis. Again, higher relative enrichments were detected in IMP and GMP in the embryos (Figure 2g). Pyrimidines were also labeled by both [U - ^{13}C]glucose and [γ - ^{15}N]glutamine, but with less consistent differences between embryo and placenta (Supplemental figure 2g,h). Overall, the data indicate rapid metabolism during midgestation, including prominent utilization of maternal glucose and glutamine for embryonic purines, and distinct patterns of metabolic labeling between embryo and placenta.”

3. The time-resolved tracing experiment (figure 3) is used to claim that glucose metabolism in embryo drifts, while in placenta it remains constant or follows an inverted trend. This is solid, but it should be noted that the ^{13}C labeling of the two compartments converge to similar values (figure 3c-e). Intuitively, this would indicate that their metabolic activity becomes similar during gestation, in sharp contrast with the claimed divergence of the embryo. Please clarify.

RESPONSE: We appreciate the comment. With respect to labeling of TCA cycle intermediates, these features were highly divergent on gd9.5 (Figure 3a), with 3-4-fold differences in all metrics of labeling relative to the precursor pyruvate (Figure 3c, Supplemental Figure 3c). TCA cycle labeling does indeed converge between the placenta and embryo over a time period coinciding with increased oxygen delivery in the embryo. However, over this same time period, metabolism begins to diverge among the embryonic organs, an important observation that documents metabolic compartmentation even within the embryo (Figure 3e). We edited the text to clarify what is divergent, convergent and compartmentalized with respect to TCA cycle labeling during midgestation. We also point out that the metabolomics data (Figure 1d and Supplemental figure 1a) reveal many other metabolic differences between the placenta and embryo that do not converge.

4. The use and relevance of the hypomorphic LIPT1 mutation require more explanations. First, disrupting LIPT1 has catastrophic consequences that reach well beyond glucose oxidation. The question is whether this experiment is specific enough to claim that glucose oxidation is essential. I would argue that inhibiting ETC or preventing pyruvate transport in mitochondria would have been a much more surgical intervention to prove the relevance of embryonic glucose oxidation during midgestation. Second, there is no data to prove that placental metabolism is not – or less - affected by the LIPT1 mutation. Supp figure 4d suggests the opposite. This seems an important control to claim compartmentalization of glucose oxidation, or that placenta function is normal (line 37 of the abstract).

RESPONSE: We analyzed the LIPT1 knock-in model to determine whether the infusion method was sensitive enough to detect metabolic perturbation in utero, and to place the demise of these mutant embryos in context of metabolic changes in the embryo between gd9.5 and gd12.5. The Reviewer is right that the N44S mutation in LIPT1 disrupts multiple aspects of mitochondrial metabolism by reducing the activity of 2-ketoacid dehydrogenases. We edited the text to avoid ascribing all the developmental defects to failed glucose oxidation. We chose this mutation because of its direct relevance to human disease – indeed, we reported it in one of our patients⁷. This hypomorphic mutation is arguably more relevant to human disease than the exon deletions typically used to create null mutations in mice. Knocking out a component of the ETC would not provide a more specific approach to inhibit glucose oxidation; on the contrary, this would impair metabolism of glucose, amino acids and fatty acids, as has been documented extensively in patients with ETC defects. The revision also includes new data demonstrating glutamine oxidation during mid-gestation (Figure 3f). This pathway involves α -ketoglutarate dehydrogenase, which is also lipoylated. This new observation increases the relevance of impairing multiple 2-ketoacid dehydrogenases to interrupt embryonic metabolism.

*To address the Reviewer's concern, we now justify using the N44S knock-in model in light of the more complete picture of fuel oxidation in embryos provided by the new data. Regarding the concern that LIPT1 deficiency has catastrophic consequences: this is true – homozygosity for the N44S mutation is lethal by gd11.5. But the same could be said for null mutations in *Pdha1* (PMID: 11708858), *MPC2* (PMID:24910426), *DLD* (PMID: 9405644) and *OGDH* (IMPC website), all of which are embryonically lethal in the homozygous state. Now that we can assess oxidation of different fuels in different organs, it will be interesting to determine how each of these mutations impairs organ-specific metabolism at mid-gestation. But analyzing these other models would not materially alter the primary conclusion that the method detects the embryo's failure to induce fuel oxidation and allows us to determine when after this failure the demise occurs. Additional information*

about the metabolic and developmental specificity of this mutation, and the specific links to the human phenotype, are presented in the response to Reviewer 3.

We do not argue that placental metabolism is unaffected by LIPT1 deficiency, only that placental changes are insufficient to explain the embryonic defects. As we pointed out in the paper, N44S homozygosity does impact placental metabolism, but the changes are less prominent than what we observe in the embryos. This applies to changes in metabolite abundance (Figure 4a,b, Supplemental Figure 4a) and labeling of TCA cycle intermediates (Figure 4c, Supplemental Figure 4f). The weights of the homozygous mutant placentas are also somewhat lower than the wild-types and heterozygotes (Supplemental Figure 4b). However, the key point is whether the changes observed in the embryo can be explained by the changes observed in the placenta. We do not think they can. First, the expression of genes related to placental development are largely preserved in the mutants (Supplemental Figure 4e). Second and more importantly, we examined the transfer of two different nutrients, glucose and glutamine, from the maternal circulation across the placenta and into the embryo. Both were preserved (Supplemental Figure 4c,d). Because data throughout the paper argue that a great deal of embryonic metabolism occurs locally, we conclude that embryonic demise is related to a failure of LIPT1-dependent reactions in the embryo rather than altered placental metabolism.

New text starting at line 185:

“*Lipt1*^{N44S/N44S} placentas had no defects in taking up [U-¹³C]glucose or [U-¹³C]glutamine from the maternal circulation or transferring the label to the embryos (Supplemental Figure 4c,d). Placental differentiation markers⁸ were largely conserved between healthy and *Lipt1*^{N44S/N44S} placentas (Supplemental Figure 4e). From this, we conclude that although LIPT1 deficiency alters placental metabolism, placental dysfunction is not the primary cause of lethality in the *Lipt1*^{N44S/N44S} embryos.”

Minor points:

1. Figure 1d: I am puzzled by the normalization of metabolomics data. If the data is autoscaled, there is no reason of using log2 on top of z-scores. In fact, it skews the data in favor of subtle changes.

RESPONSE: We apologize for the confusion. We used the online tool Metaboanalyst 5.0 to log transform and autoscale the data prior to generating PCA plots and heatmaps. Statistical significance for individual metabolites was determined by t-test or ANOVA on the log-transformed data. We did not use z-scores.

2. Lines 93-95: the lower enrichment of G6P or F6P cannot be justified by exchange with trioses in lower glycolysis. Beyond generating unlabeled G6P/F6P, it would also produce m+4 molecules with equal likelihood. Since m+4 is not detectable (Supp Figure 2b), the hypothesis is wrong.

*RESPONSE: Please note that the question refers to infusions with [1,2-¹³C]glucose, which were removed from the paper. Nevertheless, we would like to address the comment, which raises an important point. We respectfully disagree with the Reviewer's interpretation. When [1,2-¹³C]glucose is metabolized through glycolysis, only half of the resulting three-carbon intermediates contain ¹³C. But the unlabeled three-carbon intermediates arise both from [1,2-¹³C]glucose and from other unlabeled sources. Therefore the overall enrichment in three-carbon intermediates is low. As shown in the left panel below for the Reveiwewer, enrichment in pyruvate is only around 10% in the placenta. The likelihood that G6P m+4 will arise is the square of the fractional enrichment in the three carbon pool, in this case about 1% (0.1 * 0.1 = 0.01). This is similar to the measured levels of m+4 G6P shown in the right panel below. Using similar logic, the likelihood of producing an unlabeled G6P under identical conditions is 0.9 * 0.9 = 0.81, much higher than the likelihood of producing m+4 G6P. As stated above, we write that glycogenolysis, gluconeogenesis and other pathways could all contribute to labeling differences between placenta and embryo.*

3. Lines 128 and following: the ratio “Pyruvate enrichment / citrate enrichment” does not inform on pyruvate oxidation (i.e. on how much of the pyruvate is oxidized), but on how much citrate originated from pyruvate. It should not be used as a proxy for the amount of pyruvate that is oxidized in mitochondria.

RESPONSE: We agree. We did not argue that the ratio reports the amount of pyruvate that is oxidized. We use the ratio to report the how much citrate originates from pyruvate, and to report the route by which pyruvate enters the TCA cycle (e.g. via PDH). This has implications for the sources of mitochondrial acetyl-CoA. To avoid confusion, we revised the statement:

New text starting at line 139:

“The citrate m+2/pyruvate m+3 ratio reports transfer of labeled 2-carbon units from glucose to citrate via pyruvate dehydrogenase (PDH), while the citrate m+3/pyruvate m+3 ratio reports transfer of labeled 3-carbon units via pyruvate carboxylase (PC) (Figure 3c, Supplemental Figure 3c). In both tissues on all days, citrate m+2/pyruvate m+3 exceeds citrate m+3/pyruvate m+3, indicating that pyruvate enters the TCA cycle predominantly by PDH.

4. Line 192: “The data indicate that lipoylation facilitates a precisely-timed change in mitochondrial metabolism required for development past gd10.5.” reads like a change in lipoylation drives the metabolic changes in glucose and is responsible for its timing. There is no evidence for this. Timing could be simply triggered by oxygenation. Please correct.”

RESPONSE: We thank the Reviewer for pointing this out. We agree that lipoylation is not solely responsible for these metabolic changes, particularly over a period where oxygenation is also increasing. Lipoylation is a post-translational modification that positively regulates the activity of PDH and other 2-ketoacid dehydrogenases in the mitochondria. It is required for substrate transfer during catalysis. This is why impaired lipoylation interferes with the increase in TCA cycle labeling that occurs during mid-gestation in response to improved oxygen delivery and perhaps other factors. To clarify this point, we edited the text:

New text starting at line 230:

“We find that LIPT1 is required for precisely-timed changes in mitochondrial metabolism required for development past gd10.5; LIPT1 mutants persist for about one day after TCA cycle labeling increases in wild-type counterparts, and then die.”

Referee #2 (Remarks to the Author):

The manuscript by Solmonson et al. entitled “Compartmentalized glucose oxidation supports midgestation mammalian development” describes a novel approach to directly assess “evolving” metabolic transitions in utero in model organisms like mice. Isotope tracing and metabolomics identify evolving metabolic programs in both the placenta and embryo over gestational days (gd)10.5 to 13.5. Interestingly, gd10.5 to gd11.5 mark a period of significant divergence between placenta and embryo proper. For example, glucose carbons’ contribution to the TCA cycle increases through midgestation in the embryo but not the placenta. Subsequently, the embryo contains organ specific (e.g. heart, brain, liver) metabolic programs that include pyruvate oxidation in some cases. Finally, the authors describe a lipoyltransferase-1 (LIPT1) mutant strain of mice and demonstrate that this results in decreased glucose oxidation in both embryo and placenta leading to

lethality at gd11.5. These findings serve as a functional confirmation of the importance enhanced glucose oxidation and other fuels at midgestation. Overall, the methods described are highly innovative and dependent on significant surgical skills, precise metabolomics, and metabolic labeling approaches. I recommend acceptance of the paper upon resolution of the following concerns:

RESPONSE: We thank the Referee for the supportive comments.

1. Data displayed in Figure 1d indicates that placenta at gd11.5 have a striking decrease in a large number of metabolites that resolves just one day later. Can the authors account for this observation?

RESPONSE: We believe the Reviewer is referring to the placental metabolites that appear blue on the heatmap at gd10.5 in the placenta and are significantly increased by gd11.5. We were intrigued by this as well and explored these metabolites further. Most of these metabolites come from a single pathway: the urea cycle. This pathway has been implicated in placenta biology for its role in nitric oxide synthesis and polyamine synthesis, both of which support placental angiogenesis and placental-fetal blood flow. Arginine deficiency is associated with pre-eclampsia in animal models and reduced placental eNOS abundance has been observed in pre-eclamptic women⁴. It is possible that the abrupt but transient increase in these metabolites reflects a brief mismatch between activation of the pathway and utilization of the metabolites. This would be interesting to study further. We comment on this in the description of our metabolomics data and included the figure below as Supplemental figure 1e. Text describing this observation starts on line 85.

2. From gd9.5 to gd12.5, the embryos exhibit more dramatic increases in mass than the placenta. Do the authors believe that the distinct metabolic programs of these two tissues support this phenotype?

RESPONSE: We believe that the increase in oxygen and nutrients delivered to the embryo during midgestation substantially increases its growth rate because prior to this stage nutrients and oxygen are limited by diffusion. The placenta is partially oxygenated already by gd9.5. We also note that many metabolites that accumulate in the embryo on gd11.5 are biosynthetic intermediates, and these are likely relevant to rapid growth of the embryo. We comment on this in the manuscript starting at line 218. Moreover, although the placenta needs to grow to keep pace with the embryo's demand for nutrients, fuel consumption by the placenta is likely limited in order to provide sufficient nutrients to the embryo.

3. Later in the manuscript, the authors rely on transcriptomics to examine glucose oxidation in fetal heart and brain, which revealed increased ETC-related transcripts particularly in the heart. By the same argument, could transcriptomics in the placenta versus the embryo explain the dilution by placental labeled glucose-6-phosphate due to glycogenolysis, gluconeogenesis, and other pathways? While RNA abundance can't explain everything, of course, it could also be another way to examine this issue as described in Figure 3f.

RESPONSE: We thank the Reviewer for this suggestion. We looked specifically at expression of genes that would contribute to an unlabeled G6P pool throughout midgestation in the placenta and embryo. Agl, PygB, PygL, and PygM are involved in glycogenolysis and Fbp1/2 are required for gluconeogenesis. Although some of these genes, particularly Pygl and Fbp2, display increased expression in the placenta relative to the embryo throughout this period (see figure below), genetic experiments to knock these enzymes out in the placenta are needed to convincingly determine the involvement of these genes in G6P metabolism. We hope to pursue these experiments in the future with placental Cre drivers.

4. The metabolomics and metabolic labeling indicate a dramatically increased expansion in the purine pool (see Figure 1f and g). The labeling scheme depicted can also detect pyrimidines? If so, why was this not performed. One wonders if this activity is to support DNA synthesis or as bioenergetics (via ADP) or both.

RESPONSE: We thank the Reviewer for this comment, which prompted us to examine pyrimidine metabolism in more depth. We initially did not focus on pyrimidines because their levels did not increase as prominently as the purines in the embryo. The expanding purine pool makes it easier to conclude that increased ^{13}C labeling in the embryo reflects enhanced biosynthesis. In pyrimidines, infusions with either $[\text{U}-^{13}\text{C}]$ glucose or $[\gamma\text{-}^{15}\text{N}]$ glutamine revealed substantial labeling in both the embryo and the placenta (Supplemental Figure 2g,h), indicating that these metabolites are also turning over rapidly. However, unlike purines, pyrimidine labeling was not consistently elevated in the embryo. These new findings are described starting on line 127. We agree that nucleotide synthesis likely contributes to both nucleic acid synthesis and bioenergetics. The involvement of guanosine nucleotides in addition to ADP strongly suggests that bioenergetic demands are not the sole reason for these pathways becoming activated.

Referee #3 (Remarks to the Author):

This is an interesting study with the ambitious goal of elucidating the metabolism of placenta and embryo during a key phase of organogenesis and embryonic development using mice as model system. The authors found that around gd10.5-11.5 the metabolism, specifically glucose metabolism, starts to compartmentalise between embryo and placenta. The required methodologies were challenging and involved in vivo assessment of metabolic activity, infusion of isotope-labeled metabolites, and testing of a mouse strain deficient in glucose oxidation.

The labeled metabolite tracing experiments with wild-type mice are mostly clear and convincing (with a few comments pointed out below). But I'm not convinced that the Lipt1N44S mice are a suitable model to study metabolism at gd10.5 when homozygotes die at the gd10.5-gd11.5 transition. The authors need to clarify and show that results are not confounded by the imminent death of embryos (specific concerns are pointed out further below).

Detailed comments:

1. Figure 1

Line 68: "Unsupervised clustering of metabolomics data indicated that placenta and embryo differ from each other on all days examined (Figure 1c,d).

Please clarify this statement because the PCA plot (Figure 1c) disagrees with this statement. Metabolites in the E13.5 placenta samples cluster inside E12.5 placenta samples. There is a large spread between samples at E12.5, which would suggest that E13.5 placentas are similar.

RESPONSE: We thank the Reviewer for the comment and apologize for the lack of clarity. We meant to say that on all days, the samples from the embryos are more similar to each other than the samples from the placentas, and that there appears to be a transition in each tissue between gd10.5 and gd11.5. This is perhaps best observed on the heatmap in Figure 1d. We have clarified this statement in line 77:

“Placenta and embryo metabolomics differ throughout midgestation, as expected given their divergent cellular composition and functions (Figure 1c,d, Supplemental figure 1a). In both tissues, gd10.5 was metabolically distinct from subsequent days, indicating transitions between gd10.5 and gd11.5 (Fig. 1d, Supplemental figure 1a). These transitions were largely private, with most metabolites changing in one tissue but not the other (Figure 1e,f Supplemental Fig. 1b-d)”

2. Figure 2

Supplemental figure 2a is helpful to understand the glucose labelling results. I suggest putting it into the main text and swapping it with the current Figure 2a which can be moved to the supplement (the procedure is described in the text).

RESPONSE: We changed the presentation of the labeling data substantially in response to Referee 1's comments. This involved taking out the data using the [1,2-13C]glucose tracer and simplifying the presentation. The main figure now includes a high-level illustration to explain the origin of the labeling features most central to the key data (Figure 2d). The illustration of the experimental technique still seems to fit, so we left it where it was. We will move this figure to the supplement if the space is needed.

3. Infusion and serial dissection of conceptuses over 3 hours (Figure 2 and page 4, line 80 onwards): The idea of doing a kinetic analysis by serially dissecting conceptuses over a time while infusing labelled metabolites is good, however, appropriate controls are lacking. It is possible that the longer the mother is under anesthesia with repeated exposure to the procedure that maternal metabolism might be affected. Controls would be (i) maintaining the anesthesia and infusion over 3 hours but only dissecting once at the end of the 3 hours; (ii) the effect of the anaesthetic can be assessed by dissecting control litters at the same embryonic stage in which the mother is culled before dissection.

RESPONSE: We apologize for not clarifying the procedure better and are thankful to the Reviewer for bringing this to our attention. We performed infusions to obtain kinetic data during serial C-sections, and separate infusions in which samples were acquired only at later time points. Infusions for 3-4 hours were performed as the Reviewer requested, with dissection only at the end of the experiment. Although there were differences in the infusion protocol between the serial c-sections and the 3-4 hour infusions (serial C-sections did not include an initial bolus and the infusion rate was 5 μ L/min), the overall pattern of enrichment was the same. We do not make any comparisons in the manuscript between serial C-sections and end-point infusion experiments. But the serial C-section experiments do provide time-resolved data relating to aspects of developmental metabolism that the longer experiments do not. We have updated the panels in Figure 2 and Supplemental Figure 2 to indicate which experiments correspond to which type of infusion.

4. Line 116: “Like R5P, serine and glycine were extensively labeled in the embryos, indicating the rapid production of multiple purine precursors (Figure 2k).”
Check this statement as Fig2k shows no difference in R5P between embryo and placenta.

RESPONSE: We thank the Reviewer for allowing us to clarify this statement. Although the total enrichment (sum of all labeled isotopologues) was no different for R5P (Figure 2f), the isotopologue distribution was distinct (Supplemental Figure 2b). In the embryo, the m+1, and m+3 isotopologues are lower than the placenta, however those isotopomers in the purine nucleotides trended higher than the placenta. This tells us that there was additional label incorporated into the purine nucleotide independent of the labeling in the R5P ring. We believe that this labeling comes from incorporation of glycine and formate in the de novo purine synthesis pathway.

5. LIPT1

The embryo/placenta data should be represented and discussed as Lipt1+/, Lipt1+/N44S or Lipt1N44S/N44S and not as healthy and mutant. If Lipt1+/, Lipt1+/N44S are grouped in the graphs, then colour code the data points so it is clear which represent the two different genotypes.

RESPONSE: We changed the figures related to LIPT1 to differentiate between LIPT1 +/+ and LIPT1 +/N44S in the healthy cohorts.

6. Fig. 4a: About half of the mutant embryos appear to be about 8 fold lighter (smaller) than the average weight of healthy embryos, whereas the placentas have similar weights between phenotypes. I am concerned that the metabolic data from these severely affected tiny embryos reflect general embryo demise rather than the specific effect of the mutation on metabolites especially given that homozygotes “undergo embryonic lethality at the gd10.5-gd11.5 transition” (line 164). Was any embryo lethality observed at gd10.5 in the litters used to generate these data? If so, the number of dead embryos should be stated per litter.

RESPONSE: Between gd10.5 and gd11.5 we sometimes see resorption of a conceptus which presumably is a LIPT1 homozygous mutant. However, these embryos are not used for any downstream analysis; all the metabolic data were derived from viable embryos. We also performed [U-¹³C]glucose tracing experiments in gd9.5 litters to show that modest metabolic alterations are present before gd10.5 (Figure 4d, Supplemental table 3). Additional data described below report developmental features at gd9.5 and gd10.5.

7. Line 164: The authors say: “Lipt1N44S/+ mice are healthy and born at the expected ratio and their metabolomic signatures are similar to Lipt1+/+ tissues at gd10.5. (Supplemental Figure 4a,b)”
The authors should improve the clarity of this paragraph and explain Supplemental Figure 4a and b better. Especially because this is a key experiment from which they drew the conclusion that both wild-types and heterozygotes are “healthy” and can be pooled for analyses. Specifically, which metabolic signatures were similar and different? Is this conclusion based on the embryo plot alone, because the separation appears less clear-cut in the placenta?

RESPONSE: The PCA plots in Supplemental Figure 4b indicate that WT/WT and WT/N44S tissues (embryo or placenta) are indistinguishable when the entire metabolome is considered. More importantly, the metabolites directly related to LIPT1 function (e.g. cis-aconitate, alpha-ketoglutarate, succinate, fumarate, malate, aspartate, lysine, KIV, KIC, KMV), all of which are perturbed in the LIPT1 homozygous embryos are no different between WT/WT and WT/N44S embryos (Figure 4a). This is the expected outcome for an autosomal recessive inborn error of metabolism. In the revision, all data in Figure 4 and Supplemental Figure 4 that involve these three genotypes are now color-coded so that readers can see which data points come from WT/WT embryos vs. WT/N44S embryos. We hope the Referee agrees that heterozygosity for the N44S allele does not induce metabolic changes in LIPT1-dependent pathways.

8. Line 167: Lipt1N44S/N44S conceptuses are small but viable at 10.5.
This should be discussed further: Are they small because they are developmentally delayed? Then the metabolic profiles at gd10.5 would be different to heterozygotes and wild-types because they resembled those of earlier conceptuses. Or are they small due to a lack of cell division, but metabolically matured like a normal gd10.5 conceptus? Since the authors did not see difference in expression markers in placentas between genotypes, and placenta weights were similar, it suggests that the Lipt1N44S/N44S placenta has developed normally.

The authors could answer this by doing a differentiation marker expression analysis (Fig S4) with embryos, as done for the placentas. It would show if the small gd10.5 embryos have similar gene expression and therefore are indeed developed to gd10.5.

On a related note, the authors should discuss what they consider the cause and effect of the observed differences in Lipt1N44S/N44S embryos. Are they small at gd10.5 because of the metabolic perturbation, i.e. cell division and maturation are impaired because of the limited energy generation via TCA cycle, etc? Or do the embryos have metabolic changes as a result of abnormal embryonic development, e.g. metabolically active cell types are underrepresented due to perturbed embryonic development?

RESPONSE: These are important questions and we thank the reviewer for raising them. We performed several kinds of experiments to better explore the relationships between the mutation and its metabolic and developmental consequences during midgestation. We performed a more complete metabolic analysis to assess the effect of the mutation on evolving metabolic activity during this period, and we undertook a much more detailed developmental analysis to understand how altered LIPT1 function impacts these processes.

For the metabolic assays, we carefully assessed the ¹³C labeling features between mutant and healthy embryos on days gd9.5 and gd10.5 (Figure 4d, Supplemental table 3). As indicated above, some suppression

of labeling is already present in the $LIPT1^{N44S/N44S}$ embryos at gd9.5, and labeling does not catch up to heterozygous and wild type embryos by gd10.5. Indeed, the labeling profile in the gd10.5 mutants is similar to the healthy embryos at gd9.5, indicating a metabolic delay (Figure 4d). As the Reviewer states, this is akin to a state of metabolic immaturity in the $LIPT1^{N44S/N44S}$ embryos.

We next recruited help from developmental biologists here at UT Southwestern to examine overall embryonic morphology as well as markers of differentiation for different tissues. This was aimed to test whether the phenotypes observed were simply a result of overall embryonic failure or whether they resulted from specific defects in select tissues. We first counted somites, which is a standard method to evaluate developmental delays. We found that although the *Lipt1* mutant embryos were smaller, they had equivalent numbers of somites, suggesting overall development was not affected (Figure 4e). We then assessed whether small embryonic size and lethality might result from vascular defects, as loss of gas/nutrient exchange would lead to similar phenotypes. Using immunofluorescence, we found that vasculogenesis (the initial formation and patterning of blood vessels) was normal and that blood vessels were present throughout the mutant embryos (Supplemental Figure 4g). We also found that vessel maturation, as assessed by the flow responsive *Connexin40*, was also normal (Supplemental Figure 4g). However, what we did observe was that both the developing brain and heart were particularly affected in *Lipt1* mutants, and significantly smaller than those in wildtype littermates (Figure 4f, Supplemental Figure 4g). Involvement of the heart and brain are interesting given the increasing prominence of glucose as a carbon source during this period.

Finally, to explore a well-characterized and quantifiable cellular lineage, we examined erythropoiesis at gd10.5. Previous work has demonstrated that mitochondrial metabolism is essential during erythroid specification⁹. Embryos of various genotypes were harvested, disaggregated, stained with CD71, TER119, and CD41 antibodies and analyzed by flow cytometry to assess the abundance of myeloid-erythroid progenitors (MEPs) and fetal erythrocytes (RBCs). We determined that *LIPT1* deficiency resulted in a 6-fold decrease in RBCs but a doubling of MEPs (Fig. 4g,h, Supplemental Fig. 4h). The accumulation of MEPs is more consistent with a block or delay in erythropoiesis than a generalized death of cells in the erythroid lineage, as increased numbers of dead cells in $LIPT1^{N44S/N44S}$ embryos were not observed. Remarkably, this phenotype is consistent with an aspect of human *LIPT1* deficiency that we had not previously appreciated. Review of the clinical history of our *LIPT1*-deficient patient reported in 2019 revealed chronic anemia that cannot be explained by common nutritional deficiencies (iron, folate, vitamin B12) and which does not involve suppression of other major hematopoietic lineages (platelet and leukocyte counts are normal). These data are in Figure 4i and Supplemental Figure 4i,j. Although detecting elevated MEPs in the patient would require an invasive procedure (i.e. bone marrow aspirate), we believe the data from the mouse and the patient are consistent with a block in the differentiation of erythroid precursors imposed by *LIPT1* deficiency.

Altogether, the data are consistent with a model in which delaying the acquisition of *LIPT1*-dependent metabolic properties delays or prevents the acquisition of developmental milestones in some but not all lineages. Different tissues clearly respond differently to $LIPT1^{N44S/N44S}$ homozygosity. Although it may not be possible to completely disentangle the relationships between metabolism and cellular composition, the simplest explanation is that the developmental problems flow from the genetically-defined metabolic defect. We discuss this point in the Conclusions section of the text and emphasize that the approach reported in the paper can be used to assess many other metabolically-derived developmental anomalies.

9. Figure 4C, D, E, F; and Supplemental Figure 4C:

Metabolite levels and expression of placental layer markers are shown as fold change from healthy placentas/embryos. How exactly was this calculated? These graphs have datasets for healthy embryos/placentas whose data points are scattered and have averages that differ from 1 (e.g. Amino adipate in Fig. 4F). A different dataset of healthy tissues must have been used for normalising and determining the fold changes. Please explain.

RESPONSE: We apologize for any confusion. We took the normalized peak areas and made them relative to the $LIPT1^{WT/WT}$ samples from that litter, to obtain a litter-specific fold change. The fold changes across all litters were then averaged and statistics was performed on all healthy ($LIPT1^{WT/WT}$, $LIPT1^{WT/N44S}$) samples compared with mutant ($LIPT1^{N44S/N44S}$). We have changed the Y-axis of these graphs to "relative abundance".

10. Conclusion

"Genetically-defined metabolic defects can result in congenital anomalies in humans, emphasizing the

importance of precise metabolic control during fetal development.” After this sentence, I expected some discussion about the relevance of this manuscript’s findings for human health and disease, but none is given. Human case reports of LIPT1 mutation exist and it might be worthwhile mentioning them? In addition, biallelic variants of the upstream genes LIPT2 and LIAS are described and lead to disease phenotypes. I would have liked to see some discussion about how the findings of this paper translate to humans, their relevance for clinical intervention, etc.

RESPONSE: We added more clinical information about these disorders in the text for Figure 4 and the Conclusions section. All known defects in lipoic acid synthesis and lipoylation impact brain development and function. It is therefore highly relevant that we detected altered brain development in the LIPT1^{N44S/N44S} embryos. The defect in erythropoiesis was surprising, because anemia is not a general feature of mitochondrial disorders, and earlier reports of LIPT1 deficiency did not comment on anemia. However, we emphasize that LIPT1 deficiency is exceptionally rare, with only five patients in the literature in addition to ours, and no patients besides ours with the N44S variant¹⁰⁻¹³. Furthermore, three of the reported LIPT1-deficient patients died within the first two weeks of life, and the other two besides ours were under 2 years old at the time of the report. Our patient’s anemia became more prominent after age 5, so it will be interesting to determine what fraction of surviving patients with defects in this pathway develop anemia.

11. “We report highly compartmentalized metabolic phenotypes in the placenta and embryo, with both tissues undergoing extensive but largely private metabolic changes between gd10.5-gd11.5. The placenta transfers glucose and other nutrients from the maternal circulation to the embryo, but also displays localized glucose turnover. Surprisingly, these aspects of placental glucose metabolism have little impact on glucose metabolism in the embryo, which is supplied almost exclusively by glucose delivered from the maternal blood.” Very interesting study but one should not be surprised that the embryo and placenta have distinct metabolic profiles as they are comprised of very distinct types and numbers of cells, with the placenta having relatively few cell types. Perhaps this point could be added to the discussion.

RESPONSE: We agree. The revision emphasizes the differences in cellular makeup between embryo and placenta starting on line 77 (see response to comment 1 above). We also added the following line to the Conclusions section (line 218):

“The metabolic differences are consistent with requirements for rapid growth, dramatically divergent cellular composition of these tissues, and evolving cellular environments.”

12. Methods

Line 239: Why were the homogenised samples for metabolomic analysis subjected to three freeze-thaw cycles with liquid nitrogen before BCA analysis and LC-MS or GC-MS? Freeze-thaw cycles are considered to be detrimental to proteins in solution and might also affect the analytes of interest. Have the authors checked the impact of the 3-times freeze-thawing on BCA assay values and the measured metabolites?

RESPONSE: The BCA analysis is used to normalize the samples to protein prior to LC-MS as opposed to attempting to normalize to the poorly-soluble protein pellet precipitated after extraction with acetonitrile and the freeze-thaw cycles. In our experience, this method provides a better approach to normalizing the extracted metabolites. Note that the purpose of this step is simply to avoid overloading the mass spectrometers, and additional internal normalizations using the total ion content (i.e. total metabolite mass) are also used to compare relative metabolite contents from one sample to another.

Minor comments:

1. ETC = electron transport chain. The abbreviation should be explained at the first appearance in the text.

RESPONSE: This has been revised.

2. Figure 1e: Abbreviation FC should be explained

RESPONSE: This has been revised.

3. Supplemental figure 3: Please provide a brief explanation what the abbreviations Ndufa, SDHb Cyc1, etc. stand for. Readers outside the field may not be immediately familiar with these genes.

RESPONSE: This has been revised.

4. Line 240: “centrifuged at top speed”. Please provide proper centrifugal force value or rpm and model of centrifuge.

RESPONSE: This has been revised.

REFERENCES

- 1 Ng, S. B. *et al.* Exome sequencing identifies the cause of a mendelian disorder. *Nat Genet* **42**, 30-35, doi:10.1038/ng.499 (2010).
- 2 Coscia, L. A. *et al.* Update on the Teratogenicity of Maternal Mycophenolate Mofetil. *J Pediatr Genet* **4**, 42-55, doi:10.1055/s-0035-1556743 (2015).
- 3 Jurecka, A., Zikanova, M., Kmoch, S. & Tylki-Szymanska, A. Adenylosuccinate lyase deficiency. *J Inherit Metab Dis* **38**, 231-242, doi:10.1007/s10545-014-9755-y (2015).
- 4 Hsu, C. N. & Tain, Y. L. Impact of Arginine Nutrition and Metabolism during Pregnancy on Offspring Outcomes. *Nutrients* **11**, doi:10.3390/nu11071452 (2019).
- 5 Prendergast, C. H. *et al.* Glucose production by the human placenta in vivo. *Placenta* **20**, 591-598, doi:10.1053/plac.1999.0419 (1999).
- 6 Tunster, S. J., Watson, E. D., Fowden, A. L. & Burton, G. J. Placental glycogen stores and fetal growth: insights from genetic mouse models. *Reproduction* **159**, R213-R235, doi:10.1530/REP-20-0007 (2020).
- 7 Ni, M. *et al.* Functional Assessment of Lipoyltransferase-1 Deficiency in Cells, Mice, and Humans. *Cell Rep* **27**, 1376-1386 e1376, doi:10.1016/j.celrep.2019.04.005 (2019).
- 8 Rhee, C. *et al.* ARID3A is required for mammalian placenta development. *Dev Biol* **422**, 83-91, doi:10.1016/j.ydbio.2016.12.003 (2017).
- 9 Liu, X. *et al.* Regulation of mitochondrial biogenesis in erythropoiesis by mTORC1-mediated protein translation. *Nat Cell Biol* **19**, 626-638, doi:10.1038/ncb3527 (2017).
- 10 Soreze, Y. *et al.* Mutations in human lipoyltransferase gene LIPT1 cause a Leigh disease with secondary deficiency for pyruvate and alpha-ketoglutarate dehydrogenase. *Orphanet J Rare Dis* **8**, 192, doi:10.1186/1750-1172-8-192 (2013).
- 11 Stowe, R. C., Sun, Q., Elsea, S. H. & Scaglia, F. LIPT1 deficiency presenting as early infantile epileptic encephalopathy, Leigh disease, and secondary pyruvate dehydrogenase complex deficiency. *Am J Med Genet A* **176**, 1184-1189, doi:10.1002/ajmg.a.38654 (2018).
- 12 Tache, V. *et al.* Lipoyltransferase 1 Gene Defect Resulting in Fatal Lactic Acidosis in Two Siblings. *Case Rep Obstet Gynecol* **2016**, 6520148, doi:10.1155/2016/6520148 (2016).
- 13 Tort, F. *et al.* Mutations in the lipoyltransferase LIPT1 gene cause a fatal disease associated with a specific lipoylation defect of the 2-ketoacid dehydrogenase complexes. *Hum Mol Genet* **23**, 1907-1915, doi:10.1093/hmg/ddt585 (2014).

Reviewer Reports on the First Revision:

Referee #1 (Remarks to the Author):

Solomonson et al. study the metabolism of mouse embryos and placenta during midgestation (day 9.5 to 13.5).

Notably, the study was done in vivo and by means of methods that allow distinguishing the two "compartments". It builds on an impressive number of tedious and technically difficult tracing experiments, which provided an unprecedented view of the gestational process. The work is divided into four parts: (I) a metabolomics analysis that highlights a separation between embryo and placenta, and its timing; (II) a ¹³C tracing analysis on intracellular glucose utilization, with a particular focus on pentose phosphate shunt and purine/pyrimidine biosynthesis; (III) ¹³C-tracing and expression analyses pivoting on mitochondrial oxidation; (IV) an analysis of animals with impaired lipoylation and, thus, mitochondrial oxidative activity. This caused embryonic lethality after gd 10.5 in spite of normal glucose uptake and breakdown to glycolysis. Overall, this study indicates fundamental and essential metabolic differences and trends between placenta and embryo during midgestation.

In the revised version, the authors addressed all my concerns. It is an inspiring landmark study!

Referee #2 (Remarks to the Author):

The authors have done a nice job responding to my concerns. The paper is now appropriate for publication by the journal.

Referee #3 (Remarks to the Author):

Thank you to the authors for addressing concerns raised. They have adequately addressed all issues except for the following more minor comments.

1st comment:

The new statement is clearer and is now highlighting the observed transition between gd10.5 and gd11.5 by referring to the appropriate figures. However, I suggest replacing the word "private" with "distinct" in line 81. I assume "distinct" was meant, but the two words are not synonymous.

There is another sentence in lines 217-218 using the word private "...with both tissues undergoing extensive but largely private changes." Distinct might be more suitable here as well.

3rd comment:

It is good that the authors performed additional experiments to check the effect of infusions for 3-4 hours and dissections only at the end of the experiment. Also, the panels in Figure 2 have been labelled clearly to indicate which data are end-point infusion and which are time-course experiments. This has added more clarity, showing that the trends are the same, confirming the general feasibility of the serial dissection experiments under sustained anaesthesia (i.e. no artefactual effects due to the repeated invasive procedure).

However, what the authors did not address was checking the effect of the anaesthetic on metabolic profiles:

"(ii) the effect of the anaesthetic can be assessed by dissecting control litters at the same

embryonic stage in which the mother is culled before dissection.”

At a minimum the authors should discuss why they consider that the anaesthetic itself would not affect the data if it is not possible to do a control experiment without infusion (as the latter likely cannot be done without anaesthetic)

6th comment:

Figure 4 has undergone a major revision and the embryo weight data is now in the Supplementary Information.

First, the authors made a mistake in Supplemental Fig 4b and accidentally showed the embryo data twice. The graph on the right-hand side needs to be replaced with the actual placenta data. Also, I still think it would be useful to provide the proportion of dead embryos that were observed, because the authors state that homozygotes “undergo embryonic lethality at the gd10.5-gd11.5 transition”. This can be done in the text by adding something like “Of the ___ embryos collected at gd10.5, ___ were resorbed.”

Also, regarding the statement “Between gd10.5 and gd11.5 we sometimes see resorption of a conceptus which presumably is a LIPT1 homozygous mutant.” The authors should do a simple check whether the numbers of surviving embryos of each genotype deviate from the Mendelian ratio (i.e. what are the proportions of each genotype?) and provide this information in the manuscript. A statistically significant lack of homozygotes would substantiate the assumption that lethality affects primarily the homozygotes.

The authors state “all the metabolic data were derived from viable embryos”.

So, were embryos that only weighed a few mg as is the case for 3/8 N44S/N44S embryos deemed viable or nonviable? If they were deemed to be nonviable, on what grounds were they dismissed? Embryos weighing so little can only be because they are either very developmentally delayed or dying.

The authors make the following statement in the updated manuscript (Line 233): “Embryonic demise involves delayed or defective development in some but not all tissues, including tissues like the heart with enhanced pyruvate oxidation over this gestational time frame, and erythrocytes, whose development requires mitochondrial function”. However, this statement that only some organs are affected is not supported by their data. Whilst the number of somites is the same and there is connexin present in the null embryos, these structures are much smaller. This is the same as the effect on the heart and brain, so I do not understand the statement that only some tissues, but not all, are affected based on the evidence that has been presented.

8th comment:

I agree with the conclusion that N44S/N44S embryos are metabolically immature. However, the authors need to be careful with statements such as “Different tissues clearly respond differently to LIPT1N44S/N44S homozygosity.” They have looked at vasculature, heart and brain, which all look to be smaller than they should be. Although the somite number is not affected, the size of the somites is almost certainly smaller as well.

In very general terms the statement is correct, different tissues indeed respond differently to the mutation. But the overall trend of the investigated tissues is developmental delay and slowed growth. This should be mentioned to not give the impression that totally different effects had been observed.

10th comment

I appreciate the additional information that has been provided about the functional impact of LIPT1 mutation in humans.

However, a minor suggestion to improve the Conclusions would be to more clearly separate statements about humans and mice. Especially in the 2 sentences of the Conclusions indicated below. The first talks about human LIPT1 and the following sentence switches to mouse, talking about gd10.5 etc. I suggest you emphasize that the following statements are about mice.

“LIPT1 activates multiple enzymes responsible for providing respiratory substrates to the TCA

cycle, and human LIPT1 deficiency results in developmental anomalies in oxidative organs including the brain. We find that LIPT1 is required for precisely-timed changes in mitochondrial metabolism required for development past gd10.5; LIPT1 mutants persist for about one day after TCA cycle labeling increases in wild-type counterparts, and then die.”

Minor comments:

All have been fixed except

1. Abbreviation ETC is not explained. I understand it is a widely used abbreviation for electron transport chain but still it should be explained at the first occurrence in the text which is in line 150.
2. I cannot find information about Ndufa, SDHb, etc. in the manuscript and neither in the legend of Supplemental Fig. 3. Please add which proteins these genes encode in the figure legend.

Author Rebuttals to First Revision:

Point-by-point response to reviewer #3

We would ask that you specifically address points #3 and #6 of reviewer #3, and to provide a point-by-point response to their report with the reformatted version of this manuscript.

Referee #3 (Remarks to the Author):

Thank you to the authors for addressing concerns raised. They have adequately addressed all issues except for the following more minor comments.

1st comment:

The new statement is clearer and is now highlighting the observed transition between gd10.5 and gd11.5 by referring to the appropriate figures. However, I suggest replacing the word “private” with “distinct” in line 81. I assume “distinct” was meant, but the two words are not synonymous.

There is another sentence in lines 217-218 using the word private “...with both tissues undergoing extensive but largely private changes.” Distinct might be more suitable here as well.

RESPONSE: This change has been made.

3rd comment:

It is good that the authors performed additional experiments to check the effect of infusions for 3-4 hours and dissections only at the end of the experiment. Also, the panels in Figure 2 have been labelled clearly to indicate which data are end-point infusion and which are time-course experiments. This has added more clarity, showing that the trends are the same, confirming the general feasibility of the serial dissection experiments under sustained anaesthesia (i.e. no artefactual effects due to the repeated invasive procedure. However, what the authors did not address was checking the effect of the anaesthetic on metabolic profiles: “(ii) the effect of the anaesthetic can be assessed by dissecting control litters at the same embryonic stage in which the mother is culled before dissection.”

At a minimum the authors should discuss why they consider that the anaesthetic itself would not affect the data if it is not possible to do a control experiment without infusion (as the latter likely cannot be done without anaesthetic)

RESPONSE: We agree with the Reviewer that anesthesia may impact some aspects of metabolism. However, our animal protocol does not allow us to sacrifice mice without anesthesia. We used two approved approaches for sacrificing animals in these studies: 1) isoflurane exposure followed by cervical dislocation, which was used in metabolomics studies that did not involve stable isotopes; and 2) ketamine/xylazine followed by cervical dislocation, which was used in stable isotope studies including serial C-sections. We do not make any comparisons between experiments using the two different approaches, and any comparisons that are reported in the manuscript are between tissues that were exposed to the same anesthetic. Therefore, if anesthesia induces metabolic changes, these changes should be equivalent between the tissues being compared (e.g. placenta vs. embryo; embryos on different days; different embryonic organs; etc). We added additional detail about the anesthesia used in these experiments in the Methods and state that we do not compare data acquired from mice exposed to different forms of anesthesia. We also emphasize that our evidence suggests that the effect of anesthesia is small relative to the intrinsic biological differences among the tissues. The metabolomics data (isoflurane) and infusion data (ketamine/xylazine) comparing healthy and LIPT1-mutant embryos in anesthetized dams are strong evidence in support of this assertion, because essentially every metabolomic alteration predicted to arise from LIPT1 deficiency was detected in the embryos (Fig. 4a-c, Extended Data Fig. 5c). Therefore, even if anesthesia alters the metabolic profile to some degree, this does not prevent us from detecting biologically meaningful changes.

6th comment:

Figure 4 has undergone a major revision and the embryo weight data is now in the Supplementary Information.

First, the authors made a mistake in Supplemental Fig 4b and accidentally showed the embryo data twice. The graph on the right-hand side needs to be replaced with the actual placenta data.

RESPONSE: Thank you for pointing out this error, which has now been corrected.

Also, I still think it would be useful to provide the proportion of dead embryos that were observed, because the authors state that homozygotes “undergo embryonic lethality at the gd10.5-gd11.5 transition”. This can be done in the text by adding something like “Of the ___ embryos collected at gd10.5, ___ were resorbed.”

Also, regarding the statement “Between gd10.5 and gd11.5 we sometimes see resorption of a conceptus which presumably is a LIPT1 homozygous mutant.” The authors should do a simple check whether the numbers of surviving embryos of each genotype deviate from the Mendelian ratio (i.e. what are the proportions of each genotype?) and provide this information in the manuscript. A statistically significant lack of homozygotes would substantiate the assumption that lethality affects primarily the homozygotes.

RESPONSE: We agree that these are important data points, but they were included in the original description of this mouse line¹. We included a line about this in the description of the model, and added another citation to the Ni et al. paper when referring to the data.

Line177:

“Mice homozygous for the N44S variant are detected at close to the expected Mendelian ratio at gd10.5 but absent by gd11.5, indicating embryonic lethality occurs between these days¹.”

The authors state “all the metabolic data were derived from viable embryos”.

So, were embryos that only weighed a few mg as is the case for 3/8 N44S/N44S embryos deemed viable or nonviable? If they were deemed to be nonviable, on what grounds were they dismissed? Embryos weighing so little can only be because they are either very developmentally delayed or dying.

RESPONSE: Embryos that showed hemorrhage and necrotic tissue were deemed “non-viable” and were not used for any downstream experiments. Any “viable” embryos (regardless of size) were included, and the analysis of the flow cytometry, histology, metabolomics, and stable isotope experiments that were performed using the whole litter and blinded to the genotypes. We accept that some embryos that were included in our studies may have been in the process of dying but we did not bias our experiments for or against these samples. Only a small number of samples were included in the tissue weights experiment and these samples were not used as part of any other experiment. However, the embryo weight data does accurately reflect our observations that some LIPT1^{N44S/N44S} embryos are similar in size to healthy littermates, yet these samples showed phenotypes consistent with smaller embryos. We hope that the fact that we performed many of our metabolic experiments at gd9.5 and gd10.5, with a large number of replicates, will reassure the Reviewer that our data is not derived from solely dead or dying embryos.

The authors make the following statement in the updated manuscript (Line 233): “Embryonic demise involves delayed or defective development in some but not all tissues, including tissues like the heart with enhanced pyruvate oxidation over this gestational time frame, and erythrocytes, whose development requires mitochondrial function”. However, this statement that only some organs are affected is not supported by their data. Whilst the number of somites is the same and there is connexin present in the null embryos, these structures are much smaller. This is the same as the effect on the heart and brain, so I do not understand the statement that only some tissues, but not all, are affected based on the evidence that has been presented.

RESPONSE: We understand the Reviewer’s point and edited this statement. Proper patterning in some of these other tissues may indicate that they are less affected than more oxidative organs like the heart. The accumulation of myeloid/erythroid progenitors in the setting of depleted mature erythrocytes also suggests that some cells tolerate LIPT1 deficiency during development. But we agree that it would be better not to make too strong a statement about tissues where development may be relatively preserved. We deleted the phrase “in some but not all tissues” from this sentence.

8th comment:

I agree with the conclusion that N44S/N44S embryos are metabolically immature. However, the authors need to be careful with statements such as “Different tissues clearly respond differently to LIPT1N44S/N44S

homozygosity.” They have looked at vasculature, heart and brain, which all look to be smaller than they should be. Although the somite number is not affected, the size of the somites is almost certainly smaller as well. In very general terms the statement is correct, different tissues indeed respond differently to the mutation. But the overall trend of the investigated tissues is developmental delay and slowed growth. This should be mentioned to not give the impression that totally different effects had been observed.

RESPONSE: We agree. As stated in the above response, we removed the comment that some tissues do not exhibit defective development.

10th comment

I appreciate the additional information that has been provided about the functional impact of LIPT1 mutation in humans. However, a minor suggestion to improve the Conclusions would be to more clearly separate statements about humans and mice. Especially in the 2 sentences of the Conclusions indicated below. The first talks about human LIPT1 and the following sentence switches to mouse, talking about gd10.5 etc. I suggest you emphasize that the following statements are about mice.

“LIPT1 activates multiple enzymes responsible for providing respiratory substrates to the TCA cycle, and human LIPT1 deficiency results in developmental anomalies in oxidative organs including the brain. We find that LIPT1 is required for precisely-timed changes in mitochondrial metabolism required for development past gd10.5; LIPT1 mutants persist for about one day after TCA cycle labeling increases in wild-type counterparts, and then die.”

RESPONSE: Thank you. We have edited this section for clarity.

Minor comments:

All have been fixed except

1. Abbreviation ETC is not explained. I understand it is a widely used abbreviation for electron transport chain but still it should be explained at the first occurrence in the text which is in line 150.
2. I cannot find information about Ndufa, SDHb, etc. in the manuscript and neither in the legend of Supplemental Fig. 3. Please add which proteins these genes encode in the figure legend.

RESPONSE: The first occurrence of ETC on line 154 has been revised to define the abbreviation as the Electron Transport Chain (ETC). All abbreviations have been included in Supplemental table 5.

Other minor edits:

1. We added information about the clinical study through which the LIPT1-deficient patient was recruited, as requested by the *Nature* editorial office. This information is now in the Methods.
2. We corrected a typographic error in Fig. 4e, right panel. This panel had been labeled “CD41+/cKit-,” but these cells do express c-Kit, as expected for myeloid/erythroid progenitors. This has been corrected to “CD41+/cKit+” on the panel.

Reference:

- 1 Ni, M. *et al.* Functional Assessment of Lipoyltransferase-1 Deficiency in Cells, Mice, and Humans. *Cell Rep* **27**, 1376-1386 e1376, doi:10.1016/j.celrep.2019.04.005 (2019).